# Endophenotype effect sizes support variant pathogenicity in monogenic disease susceptibility genes

Jennifer L. Halford [1,2,36], Valerie N. Morrill[1,3,36], Seung Hoan Choi [1], Sean J. Jurgens [1,4], Giorgio Melloni[5], Nicholas A. Marston[5], Lu-Chen Weng [1,3], Victor Nauffal[1], Amelia W. Hall[6], Sophia Gunn[7], Christina A. Austin-Tse[8,9,10], James P. Pirruccello [1,3], Shaan Khurshid[1,3,11], Heidi L. Rehm[1,9,10], Emelia J. Benjamin [12,13,14], Eric Boerwinkle[15], Jennifer A. Brody [16], Adolfo Correa [17], Brandon K. Fornwalt[18,19,20], Namrata Gupta[1], Christopher M. Haggerty[18,19], Stephanie Harris[3], Susan R. Heckbert [16,21], Charles C. Hong [22], Charles Kooperberg [23], Henry J. Lin [24], Ruth J. F. Loos [25,26], Braxton D. Mitchell[22,27], Alanna C. Morrison[15], Wendy Post[28], Bruce M. Psaty [16,21,29], Susan Redline [30], Kenneth M. Rice [31], Stephen S. Rich [32], Jerome I. Rotter[24], Peter F. Schnatz[33], Elsayed Z. Soliman [34], Nona Sotoodehnia[16,35], Eugene K. Wong[3,10], NHLBI Trans-Omics for Precision Medicine (TOPMed) Consortium, Marc S. Sabatine[5], Christian T. Ruff[5], Kathryn L. Lunetta [7], Patrick T. Ellinor [1,3,11,37] & Steven A. Lubitz [1,3,11,37] ✉

Accurate and efficient classification of variant pathogenicity is critical for research and clinical care. Using data from three large studies, we demonstrate that population-based associations between rare variants and quantitative endophenotypes for three monogenic diseases (low-density-lipoprotein cholesterol for familial hypercholesterolemia, electrocardiographic QTc interval for long QT syndrome, and glycosylated hemoglobin for maturity-onset diabetes of the young) provide evidence for variant pathogenicity. Effect sizes are associated with pathogenic ClinVar assertions ($P < 0.001$ for each trait) and discriminate pathogenic from non-pathogenic variants (area under the curve 0.82-0.84 across endophenotypes). An effect size threshold of $\geq 0.5$ times the endophenotype standard deviation nominates up to 35% of rare variants of uncertain significance or not in ClinVar in disease susceptibility genes with pathogenic potential. We propose that variant associations with quantitative endophenotypes for monogenic diseases can provide evidence supporting pathogenicity.

Determining the clinical significance of rare genetic variation has critical implications for research and optimal care for patients and their families[1]. Incorrect classification of genetic variation, though rare, may place patients and their relatives at risk for adverse consequences of disease, inappropriate therapies with related complications, and anxiety[2]. Insufficient evidence for pathogenicity is common and the inability to discriminate variants of uncertain significance (VUS) remains a significant barrier[3]. In clinical practice, genetic testing is

A full list of author affiliations appears at the end of the paper. ✉e-mail: slubitz@mgh.harvard.edu

frequently hampered by discovery of VUS and conflicting variant classifications between laboratories, with lack of specific evidence for pathogenicity due to the genetic heterogeneity of disease and rarity of segregation data[4–6]. Although a process of variant classification advanced by the American College of Medical Genetics (ACMG) and Association for Molecular Pathology exists[7], it is laborious and prone to adjudicator disagreement[8,9]. For example, one study across three laboratories found low concordance (Cohen $K = 0.26$) in classifying variants in *SCN5A* and *KCNH2*—two disease-causing genes for the long QT syndrome (LQTS) which can cause sudden cardiac death[10]. Nevertheless, with increasing use of genetic sequencing in both research and clinical practice there is now a rapidly growing pool of rare variants requiring adjudication.

Quantitative endophenotypes are instrumental phenotypes for genetic association and are often more easily and reliably ascertained than dichotomous disease status indicators. The emergence of large-scale human genetic sequence and phenotype data in biorepositories provides an opportunity to assess whether endophenotypes for monogenic diseases can be leveraged to enable scalable and accurate variant pathogenicity assertions. Currently, variant classification practices do not account for rare variant associations with endophenotypes for monogenic diseases. We hypothesized that quantitative endophenotypes for monogenic diseases measured at population-scale can provide evidence of variant pathogenicity.

To assess our hypothesis, we studied three monogenic diseases for which easily ascertained human-derived endophenotypes exist, and ambiguous variant classification is an established challenge[2,4,6,11]. Familial hypercholesterolemia (FH) is the most common inherited disorder in medicine, affecting about 1 in 250 individuals. FH is characterized by elevated blood levels of low-density lipoprotein cholesterol (LDL-C), which may lead to premature coronary artery disease and myocardial infarction[3]. LQTS is a common cause of arrhythmias that affects about 1 in 2000 individuals and may lead to sudden cardiac death[12]. The electrocardiographic corrected QT interval (QTc) is an easily measured endophenotype for LQTS. Maturity-onset diabetes of the young (MODY) is a monogenic form of diabetes often misdiagnosed as type 1 or type 2 diabetes that affects about 1 in 10,000 adults[13]. Glycosylated hemoglobin (HbA1c) levels are a routinely measured blood biomarker for diabetes.

We studied the relations between rare coding variants, defined here as those with a minor allele frequency (MAF) < 0.1%, in disease-causing genes for FH and MODY with LDL-C and HbA1c levels, respectively, among individuals who underwent whole exome sequencing (WES) in the UK Biobank (UKBB) and replicated findings

in the Further Cardiovascular Outcomes Research With PCSK9 Inhibition in Subjects With Elevated Risk (FOURIER) randomized controlled trial[14]. We studied the relations between rare variants in LQTS susceptibility genes and QTc intervals among individuals who underwent WES in the UKBB and replicated findings in the National Heart Lung and Blood Institute's (NHLBI) Trans-Omics for Precision Medicine (TOPMed) program[15], in which samples underwent whole genome sequencing (WGS).

In this work, we show that effect sizes are associated with pathogenic ClinVar assertions and discriminate pathogenic from non-pathogenic variants for three monogenic diseases. As such, variant associations with quantitative endophenotypes can provide evidence supporting pathogenicity.

## Results

### Sample characteristics

Our discovery analysis included multi-ancestry samples from the UKBB comprising 189,652 participants with LDL-C measurements, 33,520 with QTc measurements, and 189,741 with HbA1c measurements (Supplementary Figs. 1–3). Sample characteristics are reported in Table 1. Our replication analyses for the LDL-C and HbA1c endophenotypes included 14,038 and 12,798 samples, respectively, from the FOURIER trial. Replication analyses for the QTc endophenotype included 26,976 individuals from TOPMed (Replication sample characteristics are reported in Supplementary Table 1). A sensitivity analysis including individuals in the UKBB of European ancestry included 165,783 participants with LDL-C measurements, 28,249 with QTc measurements, and 166,335 with HbA1c measurements (European sample characteristics are reported in Supplementary Table 2).

### Variant-level effect sizes and pathogenicity category

Within definitive FH genes (*LDLR, APOB, PCSK9*)[16], we observed 3495 rare (within-sample MAF < 0.1%) coding variants in the UKBB; of these, 1443 (41.3%) were present in ClinVar with clinical pathogenicity assertions. Single variant association testing between each variant in the FH genes and LDL-C levels produced variant effect sizes that differed by pathogenicity category. Pathogenic and likely pathogenic variants exhibited the largest mean effect sizes (effect size ± standard deviation [SD]: 46.57 ± 52.58 and 44.50 ± 58.43 milligrams per deciliter [mg/dL], respectively) (Fig. 1A). For variants within definitive LQTS genes (*KCNQ1, KCNH2, SCN5A*)[16], we observed 1078 rare coding variants in the UKBB, 752 (69.8%) of which were present in ClinVar. Variant effect sizes for QTc duration differed across categories with pathogenic and likely pathogenic variants exhibiting the largest mean effect

## Table 1 | Baseline cohort characteristics by endophenotype in the UK Biobank

| Characteristic | LDL-C (mg/dL) | QTc (ms) | HbA1c (%) |
|---|---|---|---|
| Participants with a measurable endophenotype, *n* | 189,652 | 33,520 | 189,741 |
| Median endophenotype value (Q1–Q3) | 136 (114–159) | 411 (396–426) | 5.4 (5.1–5.6) |
| Male, *n* (%) | 85,197 (44.9) | 16,582 (49.5) | 85,210 (44.9) |
| European ancestry, *n* (%) | 165,783 (87.4) | 28,249 (84.3) | 166,335 (87.7) |
| Mean age, years (SD) | 57.0 (8.1) | 52.6 (5.7) | 57.0 (8.1) |
| Myocardial infarction, *n* (%) | 2995 (1.6) | 483 (1.4) | - |
| Statin usage, *n* (%) | 31,909 (16.8) | - | - |
| Median high-density lipoprotein, mg/dL (Q1–Q3) | 56.0 (46.3–64.0) | - | - |
| Heart failure, *n* (%) | - | 74 (0.2) | - |
| Beta blocker usage, *n* (%) | - | 1701 (5.1) | - |
| Calcium channel blocker usage, *n* (%) | - | 2455 (7.3) | - |
| Type 2 diabetes medication usage, *n* (%) | - | - | 7090 (3.7) |
| Mean corpuscular volume, femtoliters (SD) | - | - | 91.2 (4.5) |

NB: only select relevant characteristics for the given monogenic disease of interest are displayed.
This table includes median imputed values for select clinical covariates.

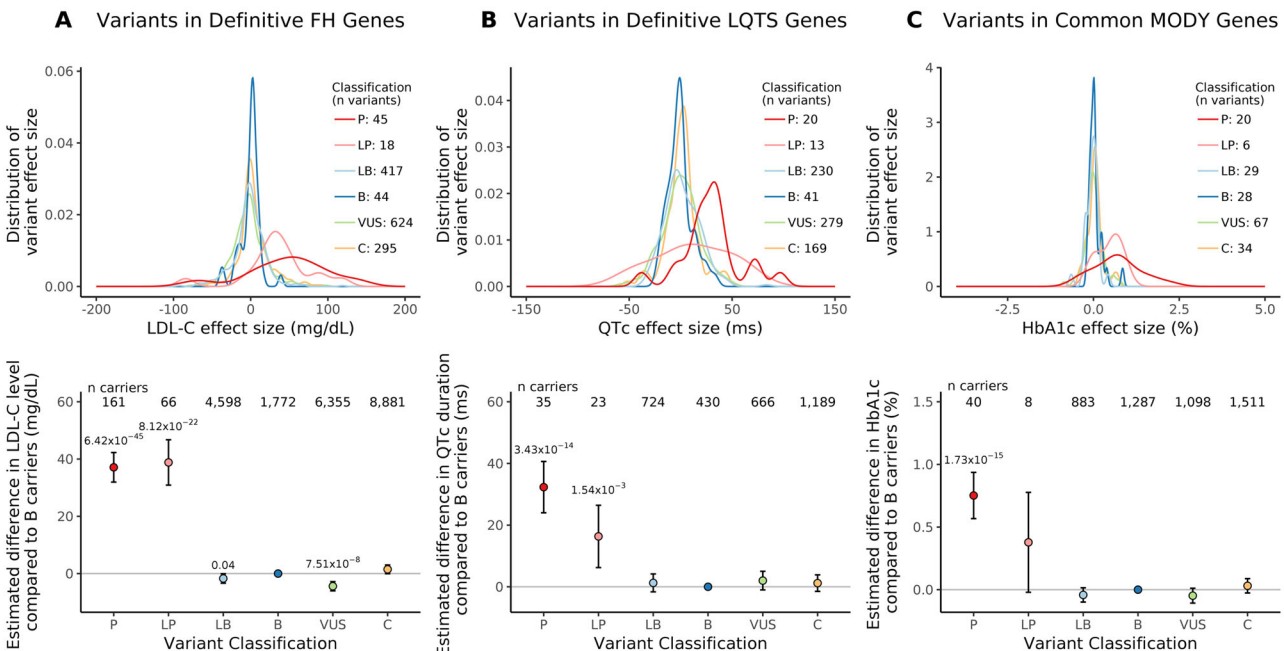

**Fig. 1 | Association between effect size and variant pathogenicity for three monogenic disease endophenotypes. A**, **B**, and **C** display data for the LDL-C, QTc, and HbA1c endophenotypes, respectively, for rare variants found in the UK Biobank. Definitive familial hypercholesterolemia (FH) genes include *LDLR*, *APOB*, *PCSK9*; definitive long-QT syndrome (LQTS) genes include *KCNQ1*, *KCNH2*, *SCN5A*; common maturity-onset diabetes of the young (MODY) genes include *HNF1A*, *HNF1B, HNF4A, GCK*. Row 1 in each panel displays the variant effect size distribution by ClinVar pathogenicity category (colored). Row 2 in each panel displays the estimated difference in endophenotype value comparing carriers of a variant in each ClinVar pathogenicity category to carriers of benign variants (circles) and 95% confidence intervals. Two-sided P values are derived from multiple linear regression model t-statistics and are annotated for variant categories with $P < 0.05$. Variant classification includes B benign, LB likely benign, LP likely pathogenic, P pathogenic, VUS variant of uncertain significance, C conflicting.

sizes ($29.58 \pm 29.19$ and $11.54 \pm 36.98$ milliseconds [ms], respectively) (Fig. 1B). For variants within the common MODY genes (*GCK, HNF1A, HNF1B, HNF4A*)[13], we observed 1052 rare coding variants in the UKBB; of these, 184 (17.5%) were present in ClinVar. Variant effect sizes for HbA1c differed across categories with pathogenic and likely pathogenic variants exhibiting the largest mean effect sizes ($0.70 \pm 0.70$ and $0.45 \pm 0.37\%$, respectively) (Fig. 1C). The distributions of effect sizes for variants tested for association with LDL-C, QTc, and HbA1c did not differ across pathogenicity categories in a control panel of hereditary cancer genes[17], which would not be expected to be associated with either LDL-C levels, QTc duration, or HbA1c percentage (Supplementary Fig. 4).

### Carrier-level effect sizes and pathogenicity category
We then assessed the estimated differences in endophenotype measurements for carriers of variants in each pathogenicity category compared to carriers of benign variants. Compared to carriers of benign variants, LDL-C levels were greater among carriers of pathogenic (difference 37.1 mg/dL, 95% CI 31.9, 42.3, $P = 6.42 \times 10^{-45}$) and likely pathogenic (difference 38.8 mg/dL, 95% CI 30.9, 46.7, $P = 8.12 \times 10^{-22}$) variants in FH genes (Fig. 1A). Similarly, QTc duration was greater among carriers of pathogenic (difference 32.3 ms, 95% CI 24.0, 40.6, $P = 3.43 \times 10^{-14}$) and likely pathogenic (difference 16.3 ms, 95% CI 6.2, 26.4, $P = 1.54 \times 10^{-3}$) variants compared to benign variant carriers (Fig. 1B). Lastly, HbA1c percentage was greater among carriers of pathogenic (difference 0.75%, 95% CI 0.57, 0.94, $P = 1.73 \times 10^{-15}$) variants compared to benign variant carriers (Fig. 1C). Carriers of likely pathogenic variants did not have significantly higher HbA1c percentage than benign variant carriers. No significant differences in LDL-C, QTc, or HbA1c measures were observed among carriers of pathogenic or likely pathogenic variants in a control set of hereditary cancer genes (Supplementary Fig. 4). In sensitivity analyses restricted to individuals of European ancestry in the UKBB ($n = 165,783$ participants for LDL-C

levels, $n = 28,249$ participants for QTc duration, $n = 166,335$ participants for HbA1c), we again observed that carriers of pathogenic variants had higher LDL-C levels, longer QTc intervals, and higher HbA1c percentages than carriers of benign variants in corresponding monogenic genes (Supplementary Fig. 5). Carriers of likely pathogenic variants had significantly higher LDL-C levels and longer QTc intervals than carriers of benign variants.

### Replication of associations between effect sizes and pathogenicity category
We observed similar relations between rare variant effect sizes for each endophenotype and variant pathogenicity categories (Supplementary Tables 6–8, Supplementary Fig. 6). Carriers of pathogenic variants had significantly greater endophenotype values compared to benign variant carriers for all endophenotypes. Carriers of likely pathogenic variants had significantly greater endophenotype values compared to benign variant carriers for LDL-C; no significant difference was observed for QTc or HbA1c.

### Variant-level effect sizes and discrimination of pathogenicity category
We then examined whether variant effect sizes can discriminate variant pathogenicity assertions. For the LDL-C endophenotype, variant effect sizes among FH genes discriminated pathogenic variants from variants classified as likely benign and benign in a logistic regression model with an area under curve (AUC) of 0.84 (95% CI 0.74, 0.93; 45 variants included) and 0.91 (95% CI 0.87, 0.96; 58 variants included) in the UKBB and FOURIER, respectively (Fig. 2A). For the QTc interval, variant effect sizes among LQTS genes discriminated pathogenic variants with an AUC of 0.83 (95% CI 0.71, 0.95; 20 variants included) and 0.79 (95% CI 0.66, 0.92; 23 variants included) in the UKBB and TOPMed, respectively (Fig. 2B). For HbA1c, variant effect sizes among MODY genes discriminated pathogenic variants with an AUC of 0.82

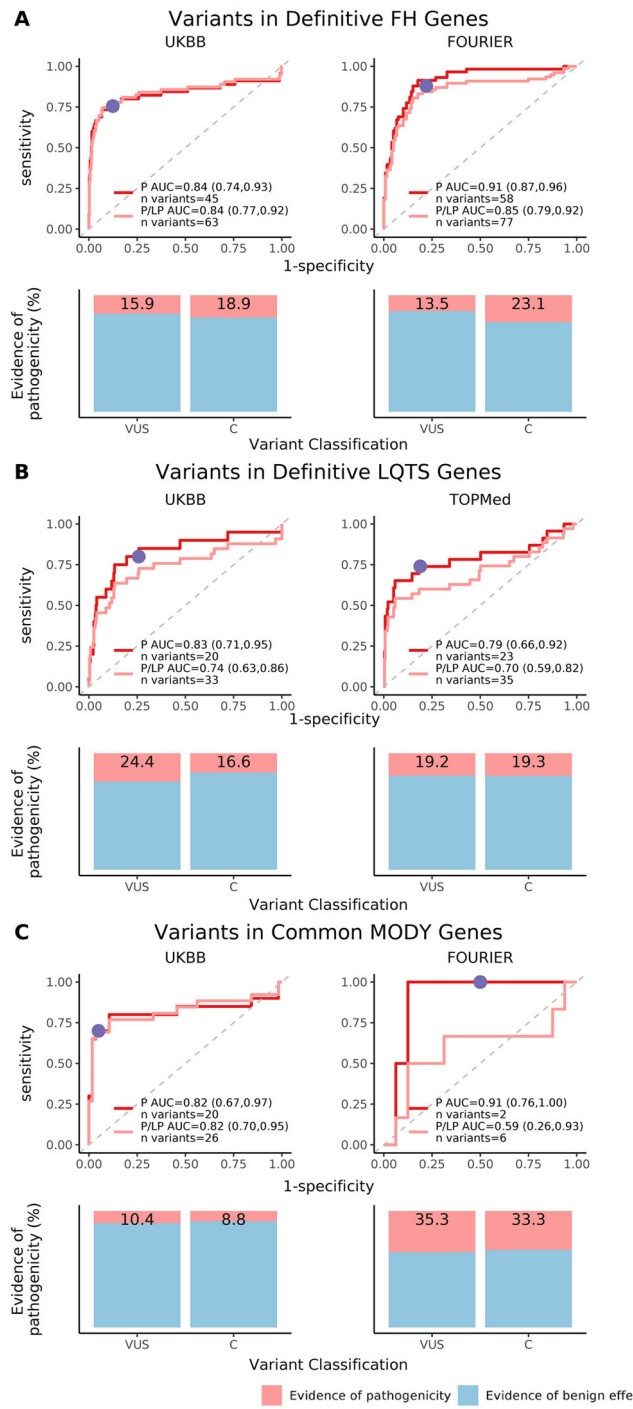

**Fig. 2 | Discrimination of variant pathogenicity by effect size and percent of variants with evidence of pathogenicity. A** This includes variants within familial hypercholesterolemia (FH) genes recommended for secondary findings return (*LDLR, APOB, PCSK9*); **B** includes variants within the long-QT syndrome (LQTS) genes recommended for which the return of a secondary variant finding is endorsed (*KCNQ1, KCNH2, SCN5A*); **C** includes variants within common maturity-onset diabetes of the young (MODY) genes (*HNF1A, HNF1B, HNF4A, GCK*). Column 1 includes variants from the UK Biobank, the primary cohort; Column 2 includes variants from FOURIER and TOPMed, the replication cohorts. Rows 1, 3, and 5 display receiver operating characteristic (ROC) curves depicting discrimination of pathogenic variants according to effect size. Variant pathogenicity is taken from ClinVar. The red curve includes "Pathogenic" variants only; the pink curve includes "Pathogenic" and "Likely pathogenic" variants combined; area under the curve (AUC) values and variant numbers are reported in the legend. Large effect size, defined as effect size greater than 0.5 standard deviations (SD) of the trait distribution, is plotted in purple on the "Pathogenic" variant ROC curve. Rows 2, 4, and 6 display variants by ClinVar category and shows the percent of variants with evidence for pathogenicity using the large effect size threshold defined in Rows 1, 3, and 5. Variants with evidence for pathogenicity, defined as having larger effect sizes than the threshold, are in pink; variants with evidence for a benign effect, defined as having smaller effect sizes than the threshold, are in light blue.

for a pathogenic or benign classification. A priori, we selected variant effect size thresholds that corresponded to 0.5 SD of the endophenotype distribution in the UKBB. As a secondary analysis, we tabulated the sensitivity, specificity, positive predictive value (PPV), and negative predictive value (NPV) of other effect size thresholds based on a range of SD thresholds (Supplementary Table 9). For LDL-C levels, a large effect size threshold corresponding to 0.5 SD of the LDL-C distribution in the UKBB was 16.7 mg/dL. In the UKBB, the large LDL-C effect size threshold had a sensitivity of 76% (95% CI 63, 88) and specificity of 88% (95% CI 85, 91) for discriminating ClinVar designated pathogenic variants from non-pathogenic variants (likely benign, and benign). PPV and NPV were calculated to be 37% (95% CI 27, 47) and 97% (95% CI 96, 99) respectively. The large effect variant threshold in the UKBB provided evidence of pathogenicity for 15.9% of VUS and 18.9% of variants with conflicting assertions in FH genes (Fig. 2A). The large effect size threshold corresponding to 0.5 SD of the QTc distribution in the UKBB was 11.9 ms, which had a sensitivity of 80% (95% CI 62, 98) and specificity of 74% (95% CI 69, 79), with PPV of 19% (95% CI 10, 27) and NPV of 98% (95% CI 96, 100). In the UKBB, this threshold provided evidence of pathogenicity for 24.4% of VUS and 16.6% of conflicting variants in the LQTS genes (Fig. 2B). For HbA1c, the large effect threshold was 0.31%, with a sensitivity of 70% (95% CI 50, 90), specificity of 95% (95% CI 89, 100), PPV of 82% (95% CI 64, 100), and NPV 90% (95% CI 82, 98); this threshold provided evidence of pathogenicity for 10.4% of VUS and 8.8% of conflicting variants in the MODY genes (Fig. 2C).

In aggregate, of the 253 VUS across FH, LQTS, and MODY genes for which large effect size thresholds provided evidence of pathogenicity, there were 244 (96.4%) missense variants, 7 (2.8%) synonymous variants, 1 (0.4%) in-frame deletion, and 1 (0.4%) in-frame insertion. We created an aggregate measure of 31 bioinformatic tools designed to predict function, which we denoted as the predicted functional impact (PFI) score which ranged from 0 to 1; higher PFI scores indicate greater predicted impact (see Methods for full detail). Of the 119 missense variants in genes associated with FH, the median PFI was 0.28 (Q1–Q3 0.09–0.41). The highest aggregate PFI was observed for variants in *LDLR* (median PFI 0.58, Q1–Q3 0.40–0.71). Of the 114 missense variants in genes associated with LQTS, the median PFI was 0.49 (Q1–Q3 0.30–0.69). Of the 11 missense variants in genes associated with MODY, the median PFI was 0.54 (Q1–Q3 0.44–0.81), with variants in *HNF1B* displaying the highest PFI of around 0.82. Detailed bioinformatic tool functional predictions are shown in

(95% CI 0.67, 0.97; 20 variants included) and 0.91 (95% CI 0.76, 1; 2 variants included) in the UKBB and FOURIER, respectively (Fig. 2C). Lower discrimination was observed when pathogenic and likely pathogenic variants were grouped together (Fig. 2). For analyses of LDL-C, 383 variants in definitive FH genes overlapped between the UKBB and FOURIER; for analyses of QTc, 439 variants in definitive LQTS genes overlapped between the UKBB and TOPMed; for analyses of HbA1c, 43 variants in common MODY genes overlapped between the UKBB and FOURIER.

**Application of an effect size threshold to provide evidence of pathogenicity**

Next, we selected a large effect size threshold which we applied to VUS or variants with conflicting assertions to document evidence

Supplementary Fig. 7; additional variant characteristics are listed in Supplementary Data 1.

There were 139 variants with conflicting assertions in ClinVar across FH, LQTS, and MODY genes for which large effect size thresholds provided evidence of pathogenicity. Of these, 112 (80.6%) were missense variants, 22 (15.8%) were synonymous variants, 2 (1.4%) were splice donor variants, 2 (1.4%) were frameshift or stop-gained variants, and 1 (0.7%) was an in-frame deletion. The PFI score was calculated for 71 variants in genes associated with FH with a median PFI of 0.63 (Q1–Q3 0.25–0.77), with the highest PFI observed in *LDLR* variants (*n* = 53, median PFI 0.76, Q1–Q3 0.57–0.80). There were 38 variants in genes associated with LQTS with a median PFI of 0.66 (Q1–Q3 0.33–0.82) and 4 variants in genes associated with MODY with a median PFI of 0.80 (Q1–Q3 0.66–0.86). Detailed bioinformatic tool functional predictions are shown in Supplementary Fig. 8; additional variant characteristics including minor allele count are listed in Supplementary Data 1.

Within the UKBB, the fraction of carriers of VUS or conflicting assertion variants with large effect size varied by endophenotype (LDL 4.7%, QTc 11.3%, and HbA1c 1.5%). In contrast, in FOURIER, a clinical trial enriched for patients with atherosclerotic cardiovascular disease, we observed a greater percentage of carriers of VUS or conflicting assertion variants with large effect sizes for the LDL (16.0%) and HbA1c (11.0%) endophenotypes (Supplementary Table 10). We submit that the increased percentage of carriers of VUS or conflicting assertion variants with large effect size for the LDL and HbA1c endophenotypes in FOURIER is consistent with a functional and clinically impactful role of such variants.

Lastly, we applied the large variant effect size threshold to novel variants that were not previously submitted to ClinVar as a screen for potential pathogenicity. In the UKBB, 438 (21.3%), 99 (30.4%), and 124 (14.3%) variants not previously submitted to ClinVar were found to have large effect size in genes associated with FH, LQTS, and MODY, respectively. Similar percentages were observed in the replication datasets (18.8%, 19.4%, and 20.1%, respectively). In total, 807 unique large effect size variants were identified across all cohorts, of which 544 (67.4%) were missense, 237 (29.4%) synonymous, 22 (2.7%) were loss-of-function (LOF; frameshift, stop-gained, or splice-altering), and the remainder with various consequences. For the 355 non-synonymous variants in FH genes, 339 (95.5%) were missense and 14 (3.9%) were LOF, with a median PFI of 0.28 (Q1–Q3 0.12–0.43). For the 106 non-synonymous variants in LQTS genes, there were 102 (96.2%) missense and 3 LOF (2.8%), with median PFI of 0.56 (Q1–Q3 0.41–0.73). There were 109 non-synonymous variants in the MODY genes, with 103 (94.5%) missense and 5 (4.6%) LOF, and median PFI of 0.63 (Q1–Q3 0.48–0.79). Additional details for the non-synonymous variants, including minor allele count, are provided in Supplementary Data 2.

As expected, we observed that the percentage of variants with large effect sizes was greatest for loss-of-function variants (38%), followed by missense/indels (22%), and synonymous variants (18%; Supplementary Table 11, Supplementary Fig. 9). The pattern of large effect size variants stratified by variant consequence was similar when we used an effect size threshold of one standard deviation of the endophenotype distribution. A total of 2261 unique synonymous variants were analyzed using SpliceAI, with 68 (3.0%) of these demonstrating some probability of being splice-altering with delta score > 0.2 (Supplementary Table 12).

## Discussion

We observed that population-level associations between genetic variants and readily ascertainable endophenotypes for monogenic diseases are informative for classifying variant pathogenicity. Specifically, rare variant effect sizes derived from association testing with LDL-C levels, electrocardiographic QTc duration, and HbA1c levels discriminated pathogenic from non-pathogenic variants in susceptibility genes for FH, LQTS, and MODY, respectively. A large variant effect size threshold provided evidence for pathogenicity for up to 35% of variants previously classified as VUS or with conflicting classifications in ClinVar. Additionally, up to 30% of variants without ClinVar assertions had large effect sizes, providing potential for pathogenicity for variants not previously subjected to rigorous clinical assertion processes. As expected, a higher proportion of loss-of-function variants had large effect sizes compared to missense/indels and synonymous variants. Similar proportions were observed for missense/indels and synonymous variants, which we submit likely reflects substantial variability in the functional role of such variants.

Our findings have two main implications. First, quantitative endophenotypes for monogenic diseases that are measurable in large scale population-based datasets may be leveraged to infer rare variant pathogenicity. We demonstrated that effect sizes for rare variants in FH, LQTS, and MODY monogenic disease susceptibility genes with corresponding endophenotypes are associated with variant pathogenicity in three large studies. Single-variant association testing in biobanks has become a widely used tool to explore the relations between rare variants and phenotypes of interest, yet it has primarily been used for genetic discovery purposes[18,19]. Our findings suggest that referencing effect sizes from large-scale rare variant association testing may have applications beyond variant or gene discovery, aiding in variant classification. The three diseases studied have precisely defined endophenotypes which are heritable, associated with the mechanism of disease, and included in the diagnostic criteria for the disease. We anticipate that using endophenotype effect sizes to aid in ascertaining variant pathogenicity will be most effectively applied to diseases with readily measurable endophenotypes in which the endophenotype is highly correlated with disease status, including other cardiac diseases such as aortic aneurysm and various cardiomyopathies[20,21]. We hypothesize that the application of endophenotype effect sizes for inferring pathogenicity may also be relevant for diseases in which endophenotypes complement diagnostic criteria[22]. Further studies are required to characterize the potential applications of the described approach and define criteria for clinically informative endophenotypes.

Second, variant effect sizes can be used as rapid and scalable discriminators of variant pathogenicity to help resolve variants with uncertain significance or conflicting classifications. Further, variant effect sizes may be useful in screening for potential pathogenicity of novel variants. The traditional variant classification process, which relies on familial segregation data and functional studies is time-consuming, labor-intensive, and subject to uncertainty[7]. While there is a growing landscape of computational tools being deployed to predict pathogenicity through assessment of variant conservation and prediction of variant effects on protein function, these tools remain imperfect and are not informed by features specific to a given disease[23,24]. We observed that the PFI, an aggregate measure of predicted variant function, was variable and heterogeneous in relation to effect size. Current algorithms for classifying variant pathogenicity do not account for population-based genotype-phenotype associations, which can be interpreted as human-derived physiologic indicators of variant expressivity. We anticipate that a potential application may be to utilize effect size information, when available, either prior to initiating a formal variant classification process or in conjunction with existing methods. Such applications will require prospective evaluation.

The large number of novel variants discovered with sequencing efforts highlights a need for rapid and scalable methods for assessing variant pathogenicity. Indeed, many individuals in our studies carried rare coding variants in disease susceptibility genes that have not previously been classified or submitted to ClinVar with clinical pathogenicity assertions (58.7% of FH variants, 30.0% of LQTS

variants, and 81.5% of MODY variants in the UKBB). We acknowledge that effect size thresholds that provide evidence of pathogenicity may differ by endophenotype and disease. Moreover, as with most tests the appropriateness of a given effect size threshold may vary by intended use of the information, which may justify a more sensitive or specific threshold (e.g., screening individuals for potential monogenic disease risk, clinical reporting of variant pathogenicity, etc.). Greater precision in effect size threshold test characteristics will follow from larger repositories of sequence and phenotype data in the future. Further analyses are warranted to examine the prognostic implications of large-effect variation, test different effect size thresholds for screening potential pathogenicity, and discover easily ascertainable endophenotypes for other monogenic diseases to aid in variant classification.

To facilitate use of our results into potential clinical practice and research, we will make variant level association results from the present analysis publicly accessible in the Cardiovascular Disease Knowledge Portal[25]. We expect that as the number of sequenced individuals grows, large compendia of variant effect sizes with endophenotypes may help classify variants as potentially pathogenic or benign, facilitating both research and clinical variant classification.

Our work must be considered in the context of the study design. The UKBB and FOURIER studies are predominantly of European ancestry and the results of our analysis may not be generalizable to all ancestries. However, the replication of associations in TOPMed, which comprises a more diverse ancestral distribution of participants suggest the results are robust. Greater ancestral diversity anticipated with future biobank efforts will further refine the ability to resolve relations between specific variants and pathogenicity. ClinVar includes variant submissions primarily from clinical laboratories in the United States[26–28] and existing variant pathogenicity assertions may not adequately represent non-European ancestral groups. Increased availability and equity of genetic testing among diverse populations is needed. We used single time-point endophenotype measurements in our analyses; repeated measures may increase measurement precision and power, and warrant examination. Rare coding variants may be subject to imprecise effect size estimation which we anticipate will improve with larger repositories of sequence and phenotypic data over time[29]. *KCNQ1*, in which variants can cause LQTS type 1, has been associated with both autosomal dominant and recessive disease; as such, calculated effect sizes from an additive genetic model may be biased toward the null[30]. We acknowledge that variant pathogenicity is not a discrete entity, and that discrimination of variants as "pathogenic" or "benign" may belie probabilistic gradients of pathogenicity or penetrance. As such, we submit that use of effect sizes may enable more quantitative inferences of variant pathogenicity.

In conclusion, population-based genetic association testing for monogenic disease endophenotypes may enable scalable inferences that provide evidence for variant pathogenicity. Future analyses are warranted to test whether large effect size variants are associated with clinical outcomes, and whether variant effect size information can be implemented in variant pathogenicity assertion workflows.

## Methods
### Study participants
The UKBB is a large, national, prospective cohort of ~500,000 individuals with detailed medical history, electronic health record, and genetic data[31]. Participants were recruited from 22 centers across the UK between 2006 and 2010 and aged 40–69 years at recruitment. Our analysis focused on participants with whole-exome sequencing (WES) and QT intervals extracted from resting 3-lead ECGs prior to a bicycle exercise protocol ($n = 33,520$), participants with WES and LDL-C measured ($n = 189,652$), and participants with WES and HbA1c measured ($n = 189,741$). Informed consent was obtained from all participants, and the UKBB received approval from the Research Ethics

Committee (11/NW/0382). Our study was approved by the Mass General Brigham Human Research Committee and conducted using the UKBB Resource (Application 17488).

### Ascertainment of clinical measurements and covariates
Detailed WES protocols for the UKBB are available elsewhere[32]. In brief, 20X whole exome sequencing was performed using IDT xGen Exome Research Panel v1.0 including supplemental probes, reads were aligned to human genome build GRCh38, joint genotype calling and initial variant QC was performed. Procedures for procuring data are discussed in detail on the UKBB website[33–35]. Clinical exclusion criteria and covariates were ascertained using self-report, ICD-9 and ICD-10 codes, and operation codes (Supplementary Methods). The extracted QT intervals were corrected using the Bazett formula, defined as $QTc = QT/\sqrt{RR}$, for subsequent analyses[36]. LDL-C levels and other disease-relevant biomarkers including HDL-C and MCV were collected at baseline from all UKBB participants and measured by enzymatic protective selection analysis on a Beckman Coulter AU5800 device. Hemoglobin A1c was measured from baseline blood tests via HPLC analysis on a Bio-Rad VARIANT II Turbo. Diabetes medication use was ascertained via self-report.

### Genotype, variant, and sample quality control
**Genotype.** The Genotype QC protocol for the UKBB is described elsewhere[37]. In brief, genotypes were removed with total depth >200 or <10. Homozygous reference calls were removed if genotype quality was <20. Homozygous alternative calls were removed if the ratio of A1 depth + A2 depth and total depth was less than 0.9, the ratio of A2 depth and total depth was less than 0.9 or phred-scaled genotype likelihood was <20. Heterozygous calls were removed if the ratio of A1 depth + A2 depth and total depth was less than 0.9, the ratio of A2 depth and total depth was less than 0.2 or phred-scaled genotype likelihood was <20.

**Variant.** UKBB variants were removed if they were in low complexity regions, had call rates <90%, failed the Hardy Weinberg Equilibrium test ($P \leq 1.0 \times 10^{-15}$), or were monomorphic in the final dataset[38,39].

**Sample.** Duplicate individuals in the UKBB were identified with KING[40] (--duplicate) and removed if not a monozygotic twin. Genetically determined sex was calculated using high quality (MAF ≥ 0.1%, missingness ≤ 1%, Hardy Weinberg Equilibrium $P \geq 10^{-6}$) independent variants on the X chromosome, as described in detail in previous studies[37–39,41]. Samples were removed if their genetically determined sex did not match their reported sex. Samples were removed if they were outliers (outside of 8 SD from the mean)[41] for quality control metrics including heterozygosity homozygosity ratio, transition and transversion ratio, SNP and Indel ratio, and the number of singletons per sample. Genetically determined ancestry groups were defined using high-quality independent variants. We estimated five ancestry groups supervised by 1000G participants using ADMIXTURE version 1.3.0[42]. We defined a sample as a member of the ancestry group if the probability of belonging to that ancestry group was greater than 80%. We defined a sample as a member of the "undetermined ancestry group" if the probability of belonging to any ancestry group was less than 80% (Supplementary Methods). We conducted our primary analysis in a multi-ancestry cohort, followed by a sensitivity analysis restricted to samples with genetically defined European ancestry.

### Clinical trait exclusions and covariates
For our LDL-C analysis, clinical covariates included high-density lipoprotein (HDL), history of myocardial infarction, and history of statin usage. We imputed participants with incomplete data for HDL to the median HDL value. We imputed participants with incomplete data

for history of myocardial infarction and history of statin usage to no history. This resulted in a cohort of 189,652 participants in our multi-ancestry cohort in the UKBB (Supplementary Fig. 1).

For our QTc analysis, participants with Wolff-Parkinson-White Syndrome (WPW), history of pacemaker placement, 2nd or 3rd degree atrioventricular block, history of class I or class III antiarrhythmic drug usage, and/or history of digoxin usage were excluded from the analysis. ECGs with QRS duration >120 ms and/or heart rate <40 or >120 beats/minute were excluded. Clinical covariates included beta blocker use, calcium channel blocker use, history of myocardial infarction, and history of heart failure. We imputed participants with incomplete data for history of myocardial infarction and history of heart failure to no history. The above QC steps resulted in a cohort of 33,520 participants for our primary analysis in UKBB (Supplementary Fig. 2).

For our HbA1c analysis, clinical covariates included mean corpuscular volume (MCV) and self-reported diabetes medication usage including common medications such as insulin, metformin, DPP-4 inhibitors, GLP-1 receptor agonists, SGLT-2 inhibitors, sulfonylureas, and thiazolidinediones. We imputed participants with incomplete data for MCV to the median MCV value. This resulted in a cohort of 189,741 participants in our multi-ancestry cohort in the UKBB (Supplementary Fig. 3).

### Variant pathogenicity assertions reported in ClinVar
ClinVar is a public database of reported sequence variants and their relations with human phenotypes[43]. We identified variants submitted to ClinVar from clinical genetic testing laboratories with the most recent pathogenicity assertion after 2015 and downloaded entries from https://ftp.ncbi.nlm.nih.gov/pub/clinvar/ on 11/28/2020. We utilized variant pathogenicity assertions that were submitted from the clinical genetic testing laboratories for further analysis. Pathogenicity categories included "benign", "likely benign", "likely pathogenic", and "pathogenic." Variants in ClinVar classified as "conflicting" due to multiple assertions with conflicting classifications and "variants of uncertain significance" (VUS) were classified as such. Ultimately, our analysis included six clinical pathogenicity categories: pathogenic, likely pathogenic, likely benign, benign, VUS, and conflicting. We then merged the ClinVar dataset with those of the UKBB, FOURIER, and TOPMed datasets by aligning with GRCh38 positions.

### Statistical analyses
**Single variant association testing.** We first estimated the empirical kinship matrix and derived principal components of ancestry using high-quality (missingness < 10%, HWE > 0.001, MAF > 0.1) independent (pruned with a window size of 200 kb, step size of 100 kb, and r2 threshold of 0.05) variants. The kinship matrix was estimated using the "--make-rel" function in PLINK 2.0[44]. Principal components of ancestry were estimated in an unrelated subset using PCAir[45]. We then derived variant effect sizes by fitting linear mixed effects models for each variant compared to non-carriers of each variant in which we regressed the endophenotype on the variant dosage, adjusting for age, sex, clinical covariates, the first 12 principal components of ancestry, and fitting a variance component proportional to the empirical kinship matrix, as well as separate residual variances for each ancestral group. We used GENESIS version 2.14.3[46] and R version 3.6[47] and assumed an additive genetic model.

### Relations between estimated variant effect size and pathogenicity.
For our LDL-C analysis, we examined the relations between estimated variant effect sizes and clinical pathogenicity assertions for variants (1) within FH genes included in the ACMG list of genes recommended for secondary findings reporting (*LDLR, APOB, PCSK9*)[16], and (2) within a control panel comprising of genes from a commercially available hereditary cancer panel (Supplementary Table 13)[17]. For our QTc analysis, we examined the relationship between estimated variant effect

sizes and clinical pathogenicity assertions for variants in established LQTS genes. We grouped variants as those residing (1) within a list of LQTS genes included in the ACMG list of genes recommended for secondary findings reporting (*KCNQ1, KCNH2, SCN5A*)[16,48], and (2) within a control panel comprising of genes from a commercially available hereditary cancer panel (Supplementary Table 13)[17]. For our HbA1c analysis, we examined the relationship between estimated variant effect sizes and clinical pathogenicity assertions for variants in established MODY genes. We grouped variants as those residing (1) within a list of established MODY genes (*GCK, HNF1A, HNF1B, HNF4A*)[13], and (2) within a control panel comprising of genes from a commercially available hereditary cancer panel (Supplementary Table 13)[17].

We assessed variant characteristics, including effect size and minor allele count, and generated density plots to assess the distribution of variant effect size stratified by pathogenicity classification category. We next assessed the estimated difference in individual endophenotype value for carriers of variants in each pathogenicity category relative to carriers of benign (B) variants using a multiple linear regression model. In the model, we regressed individual endophenotype values against carrier status of variants in each pathogenicity category, adjusting for age, sex, clinical covariates, and first 12 principal components of ancestry. Unadjusted $P$ values corresponding to the pathogenicity category terms were reported. Analyses were conducted for each of the gene panels of interest. In instances in which statistical testing was performed, we report $P$ values that are not adjusted for multiple hypothesis testing. We considered a two-sided $P$ value of 0.05 significant. Analyses were performed using R version 3.6[47].

### Sensitivity analyses in European ancestry cohorts
We performed sensitivity analyses in a UKBB European ancestry cohort for all analyses (LDL-C $n = 165,783$, QTc $n = 28,249$, HbA1c $n = 166,335$). In single variant association testing, we used the first 4 principal components of ancestry; all other procedures remained the same.

### Replication analyses in FOURIER and TOPMed
Our LDL-C and HbA1c analyses was replicated in the Further Cardiovascular Outcomes Research With PCSK9 Inhibition in Subjects With Elevated Risk (FOURIER) trial cohort. We performed variant and sample QC as outlined above. The same clinical covariates were used; of note, all participants in FOURIER were taking statin medications. After sample and clinical trait exclusions as outlined above, 14,038 participants remained for our LDL-C replication analysis, and 12,798 participants remained for our HbA1c replication analysis in FOURIER.

Our QTc replication cohort included subjects from the NHLBI TOPMed program[49] with WGS and ECG data. The present analysis includes nine studies including the Atherosclerosis Risk in Communities study, Genetics of Cardiometabolic Health in the Amish, Mount Sinai BioMe Biobank, Cleveland Family Study, Cardiovascular Health Study, Framingham Heart Study, Jackson Heart Study, Multi-Ethnic Study of Atherosclerosis, and Women's Health Initiative. Use of the TOPMed cohort for analysis was approved under paper proposal ID 8472. Genomic data included those available in Freeze 6. Details of comprising studies are provided in the supplement. Detailed WGS and variant calling protocols for the samples in TOPMed are provided on the TOPMed website[15,49]. Briefly, 30× whole genome sequencing was performed, reads were aligned to human genome build GRCh38, joint genotype calling was performed, and initial sample QC was performed on all samples by the TOPMed Informatics Research Center. Clinical covariates including QT interval were defined using study-specific definitions. These measures were centrally collected and harmonized prior to our analysis[39]. We performed variant and sample QC as outlined above. We subsetted the TOPMed WGS dataset to coding regions, defined as exons flanked by 5 base pairs for consistency with the UKBB exome cohort[39]. After sample and clinical trait exclusions as

outlined above, 26,976 participants remained for our replication analysis in TOPMed.

## Assessment of variant effect size as a discriminator for pathogenicity

We tested whether rare (MAF < 0.1%) variant effect size discriminates pathogenic from non-pathogenic variants by fitting unadjusted logistic regression models in which we regressed the log-odds of a variant being pathogenic on the variant effect size. Only variants present in ClinVar were included in this analysis. Non-pathogenic variants included those adjudicated as likely benign and benign. In sensitivity analyses, we grouped pathogenic and likely pathogenic variants as one category and regressed on variant effect size. Analyses were conducted for each of the gene panels of interest in the primary and replication cohorts. We generated receiver operating characteristic curves and calculated AUC for each using the pROC package in R[50].

## Variant reclassification and nomination of potential pathogenic variants

Lastly, we used a large effect size threshold as a screen for pathogenicity for rare variants within genes of interest. We defined the "large effect size" threshold as effect size greater than 0.5 SD of the trait distribution in the UKBB, the primary cohort. We applied this threshold to rare variants submitted to ClinVar with VUS and conflicting assertions and calculated the proportion of variants with pathogenic potential using this method for each cohort. Variant consequences and relevant characteristics including HGVS descriptions and gnomAD minor allele frequencies were annotated using the Ensembl Variant Effect Predictor tool[51]. The percentage of carriers of variants of large effect size with ClinVar uncertain significance or conflicting assertion was calculated for each endophenotype and cohort.

We characterized the potential functional consequences of the non-synonymous variants with pathogenic potential by aggregating the output of 31 in silico prediction algorithms in dbNSFP v4.2a[52] into a PFI score. Both qualitative (SIFT, SIFT4G, Polyphen2 HDIV, Polyphen2 HVAR, LRT, MutationTaster, FATHMM, PROVEAN, MetaSVM, MetaLR, MetaRNN, M-CAP, PrimateAI, deogen2, BayesDel addAF, BayesDel noAF, ClinPred, LIST-S2, fathmm-MKL, fathmm-XF, MutationAssessor, and ALoFT) and quantitative (VEST v4.0, REVEL, MutPred, MVP, MPC, DANN, CADD, Eigen, and Eigen-PC) prediction algorithms were included. Each missense variant gained 1 point per algorithm if predicted to have a functional impact (designated as "D" for qualitative tools, "H" for MutationAssessor, and >90% for quantitative tools); additional details regarding functional impact scores and cutoffs are publicly available[52]. When algorithms did not generate a prediction for the variant of interest, they received an "NA" designation. PFI scores were then calculated for each variant by dividing the number of bioinformatic tools predicting the variant to have a functional consequence by the total number of bioinformatic tools with variant prediction information available such that scores ranged from 0 to 1, with higher values indicating greater predicted impact. Heatmaps were generated to display the in silico predictions of functional consequences of VUS and conflicting variants using ggplot2[53] in R version 3.6[47].

We also applied this threshold to rare variants not reported in ClinVar and assessed the functional consequences of non-synonymous variants using bioinformatic tools. Then, we examined the proportion of large effect size variants stratified by variant consequence, including loss-of-function (frameshift, stop-gained, and splice altering variants), missense and indels, and synonymous variants. Synonymous variants located in genes associated with each endophenotype were analyzed using SpliceAI for splice-altering potential[54]. Delta scores of 0–1 were generated, with 1 representing the highest probability of the variant being splice-altering. Lastly, we tabulated the sensitivity, specificity, positive predictive value, and negative predictive value of other effect size thresholds based on a range of SD cutoffs for each trait distribution.

## Reporting summary
Further information on research design is available in the Nature Research Reporting Summary linked to this article.

## Data availability
To facilitate use of our results into potential clinical practice and research, we will make variant level association results from the present analysis publicly accessible in the Cardiovascular Disease Knowledge Portal (https://cvd.hugeamp.org/downloads.html). Access to individual-level UK Biobank data is available to researchers through application on the UK Biobank website (https://www.ukbiobank.ac.uk). The use of UK Biobank data was performed under application number 17488. Data availability from TOPMed and FOURIER are subject to controlled access. Other datasets used in this manuscript include: the ClinVar database (https://www.ncbi.nlm.nih.gov/clinvar/) downloaded in November 2020 and the dbNSFP database v4.2a (https://sites.google.com/site/jpopgen/dbNSFP).

## Code availability
Custom code or mathematical algorithms used to generate results reported in the manuscript can be obtained from contacting the corresponding author.

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

## Acknowledgements

All funding for this work is provided in Supplementary Note 1 in Supplementary Information. UK Biobank: Use of UK Biobank data was performed under application number 17488 and was approved by the local Massachusetts General Hospital institutional review board. Trans-Omics in Precision Medicine (TOPMed) program: We gratefully acknowledge the studies and participants who provided biological samples and data for TOPMed. Further Cardiovascular Outcomes Research With PCSK9 Inhibition in Subjects With Elevated Risk (FOURIER) clinical trial: We wish to thank all participants on the FOURIER clinical trial. We would also like to acknowledge Giorgio Melloni and Nicholas Marston for facilitating our work with the FOURIER data. Clinical trial and exome sequencing was supported by Amgen.

## Author contributions

J.L.H., V.N.M, P.T.E., and S.A.L. conceived and designed the study. J.L.H, V.N.M., S.H.C., and S.J.J. performed data curation and processing for data other than the FOURIER dataset. J.L.H., V.N.M., G.M.,N.A.M. performed data curation and processing for the FOURIER dataset. J.L.H. and

V.N.M. performed statistical analyses and data visualization. K.L.L., P.T.E., and S.A.L. supervised the overall study. J.L.H., V.N.M., and S.A.L. drafted the manuscript. S.H.C., S.J.J., H.L.R., C.A.A-T. contributed critically to the analysis plan and revisions of the manuscript. J.L.H., V.N.M., S.H.C., S.J.J., G.M., N.A.M., L-C.W., V.N., A.W.H., S.G., C.A.A., J.P.P., S.K., H.L.R., E.J.B., E.B., J.A.B., A.C., B.K.F., N.G., C.M.H., S.H., S.R.H., C.C.H., C.K., H.J.L., R.J.L., B.D.M., A.C.M., W.P., B.M.P., S.R., K.M.R., S.S.R., J.I.R., P.F.S., E.Z.S., N.S., E.K.W., M.S.S., C.T.R., K.L.L., P.T.E., and S.A.L. critically revised and approved the manuscript.

## Competing interests

B.M.P. serves on the Steering Committee of the Yale Open Data Access Project funded by Johnson & Johnson. M.S.S. reports grants from Amgen, Anthos Therapeutics, AstraZeneca, Bayer, Daiichi-Sankyo, Eisai, Intarcia, IONIS, Medicines Company, MedImmune, Merck, Novartis, Pfizer, and Quark Pharmaceuticals, and consulting for Althera, Amgen, Anthos Therapeutics, AstraZeneca, Bristol-Myers Squibb, CVS Caremark, DalCor, Dr. Reddy's Laboratories, Fibrogen, IFM Therapeutics, Intarcia, MedImmune, Merck, and Novo Nordisk. C.T.R. reports grant support from Boehringer Ingelheim, Daiichi Sankyo, MedImmune, and the National Institute of Health and has received consulting fees from Bayer, Bristol Myers Squibb, Boehringer Ingelheim, Daiichi Sankyo, Janssen, MedImmune, Pfizer, Portola, and Anthos. P.T.E. receives sponsored research support from Bayer AG and IBM Health, and he has served on advisory boards or consulted for Bayer AG, MyoKardia, Quest Diagnostics, and Novartis. B.K.F. is an employee of Tempus Labs. C.M.H. receives sponsored research support from Tempus Labs and has consulted for Tempus Labs. L.-C.W. receives research support from IBM Health. S.A.L. is a full-time employee of Novartis as of 18 July 2022. S.A.L. has received sponsored research support from Bristol Myers Squibb, Pfizer, Boehringer Ingelheim, Fitbit, Medtronic, Premier, and IBM, and has consulted for Bristol Myers Squibb, Pfizer, Blackstone Life Sciences, and Invitae. The remaining authors have declared no competing interest.

## Additional information

[1]Cardiovascular Disease Initiative, Broad Institute of MIT and Harvard, Cambridge, MA, USA. [2]Department of Medicine, Massachusetts General Hospital, Boston, MA, USA. [3]Cardiovascular Research Center, Massachusetts General Hospital, Boston, MA, USA. [4]Department of Experimental Cardiology, Amsterdam UMC, Amsterdam, Netherlands. [5]TIMI Study Group, Division of Cardiovascular Medicine, Brigham and Women's Hospital, Boston, MA, USA. [6]Gene Regulation Observatory and Epigenomics Platform, Broad Institute of MIT and Harvard, Cambridge, MA, USA. [7]Department of Biostatistics, Boston University School of Public Health, Boston, MA, USA. [8]Laboratory for Molecular Medicine, Mass General Brigham Personalized Medicine, Cambridge, MA, USA. [9]Harvard Medical School, Boston, MA, USA. [10]Center for Genomic Medicine, Massachusetts General Hospital, Boston, MA, USA. [11]Demoulas Center for Cardiac Arrhythmias, Massachusetts General Hospital, Boston, MA, USA. [12]NHLBI and Boston University's Framingham Heart Study, Framingham, MA, USA. [13]Department of Medicine, Boston Medical Center, Boston University School of Medicine, Boston, MA, USA. [14]Department of Epidemiology, Boston University School of Public Health, Boston, MA, USA. [15]Human Genetics Center, Department of Epidemiology, Human Genetics, and Environmental Sciences, School of Public Health, The University of Texas Health Science Center at Houston, Houston, Texas, USA. [16]Cardiovascular Health Research Unit, Department of Medicine, University of Washington, Seattle, WA, USA. [17]Departments of Medicine, Pediatrics and Population Health Science, University of Mississippi Medical Center, Jackson, MS, USA. [18]Department of Translational Data Science and Informatics, Geisinger, Danville, PA, USA. [19]Heart Institute, Geisinger, Danville, PA, USA. [20]Department of Radiology, Geisinger, Danville, PA, USA. [21]Department of Epidemiology, University of Washington, Seattle, Washington, USA. [22]University of Maryland School of Medicine, Baltimore, Maryland, USA. [23]Division of Public Health Sciences, Fred Hutchinson Cancer Center, Seattle, WA, USA. [24]The Institute for Translational Genomics and Population Sciences, Department of Pediatrics, The Lundquist Institute for Biomedical Innovation at Harbor-UCLA Medical Center, Torrance, CA, USA. [25]The Charles Bronfman Institute for Personalized Medicine, Icahn School of Medicine at Mount Sinai, 10029 New York, NY, USA. [26]The Mindich Child Health and Development Institute, Icahn School of Medicine at Mount Sinai, 10029 New York, NY, USA. [27]Geriatrics Research and Education Clinical Center, Baltimore Veterans Administration Medical Center, Baltimore, Maryland, USA. [28]Division of Cardiology, Johns Hopkins Medicine, Baltimore, MD, USA. [29]Department of Health Systems and Population Health, University of Washington, Seattle, Washington, USA. [30]Department of Medicine, Brigham and Women's Hospital, Boston, MA, USA. [31]Department of Biostatistics, University of Washington, Seattle, WA, USA. [32]Center for Public Health Genomics, Department of Public Health Sciences, University of Virginia, Charlottesville, VA, USA. [33]Department of ObGyn, The Reading Hospital of Tower Health, Reading, PA, USA. [34]Epidemiological Cardiology Research Center, Wake Forest School of Medicine, Winston Salem, NC, USA. [35]Division of Cardiology, Department of Medicine, University of Washington, Seattle, WA, USA. [36]These authors contributed equally: Jennifer L. Halford, Valerie N. Morrill. [37]These authors jointly supervised this work: Patrick T. Ellinor, Steven A. Lubitz. ✉e-mail: slubitz@mgh.harvard.edu

# NHLBI Trans-Omics for Precision Medicine (TOPMed) Consortium

Seung Hoan Choi [1], Lu-Chen Weng [1,3], Emelia J. Benjamin [12,13,14], Eric Boerwinkle[15], Jennifer A. Brody [16], Adolfo Correa [17], Susan R. Heckbert [16,21], Charles C. Hong [22], Charles Kooperberg [23], Henry J. Lin [24], Ruth J. F. Loos [25,26], Braxton D. Mitchell[22,27], Alanna C. Morrison [15], Wendy Post[28], Bruce M. Psaty [16,21,29], Susan Redline [30], Kenneth M. Rice [31], Stephen S. Rich [32], Jerome I. Rotter[24], Peter F. Schnatz[33], Elsayed Z. Soliman [34], Nona Sotoodehnia[16,35], Kathryn L. Lunetta [7], Patrick T. Ellinor [1,3,11,37] & Steven A. Lubitz [1,3,11,37] ✉

A full list of members and their affiliations appears in the Supplementary Information.

