## [Peer Review File · Nature Communications]

Endophenotype Effect Sizes Support Variant Pathogenicity in Monogenic Disease Susceptibility GenesREVIEWER COMMENTS

Reviewer #1 (Remarks to the Author):

In their manuscript, the authors describe a novel approach to utilize endophenotype effect sizes to classify variants as pathogenic. For this, they use data for endophenotypes for three monogenic diseases in the UKBB data as well as matching replication cohorts. Given the large proportion of variants with unresolved pathogenicity, the approach is very interesting and potentially useful.

Based on the currently presented material, however, the value of the approach is not yet clear because some methods are not described in sufficient detail, and statistical analyses should be complemented.

First, in general, thresholds applied in quality control and data analysis need to be justified since, in most cases, no generally accepted standards exist, and examples for that are given below.

Second, it is unclear how the authors handle the multiple testing situation.

Third, whenever possible, confidence intervals should be provided in addition to effect estimates. Specifically, providing confidence intervals for the estimates of the AUC, sensitivity and specificity should be given to facilitate the evaluation of the estimated values. For example, for LDL-C, the chosen threshold leads to a sensitivity of 75%, which is associated with a 95% confidence interval of about 60% to 87% in this sample size, and it then needs to be discussed whether these values are practically relevant.

Fourth, diagnostic values should be complemented by positive and negative predictive values to estimate the probability for a variant to be pathogenic given the effect size exceeds the given threshold. For the example of LDL-C, the PPV is roughly estimated to be below 40% if the a priori probability is taken from the proportion of pathogenic variants in all variants excluding those classified as VUS or C. The interpretation and the discussion of the results then needs to take these values into account.

Fifth, much of the interpretation of the results focuses on the proportion of as yet unresolved variants to be classified as pathogenic given the defined thresholds. It should be made clear why this is an important result. The proportion of variants classified as pathogenic could easily be varied by using other thresholds, and the thresholds used are not optimized with regard to classification accuracy. Also, it is unclear how the algorithm including the defined thresholds would perform in currently unresolved variants, given that these are likely to systematically differ from those variants for which the pathogenicity is known.

Finally, many statements in the results interpretation and discussion need to be toned down, and examples for that are given below.

Introduction

I 102: It is stated that “quantitative endophenotypes are inherently more powerful phenotypes for genetic association than dichotomous disease status indicators”. In general, it is true that the analysis of

quantitative traits is more powerful than the analysis of qualitative traits. However, this cannot directly be translated to endophenotypes versus binary disease states, because this additionally depends on the strength of the association between the endophenotype and the disease state.

I 111: Please discuss whether the selected monogenic diseases are representative for others; does the selection of those diseases for which easily ascertained human-derived endophenotypes exist restrict the generalizability of the results?

Results

I 226: “The large effect variant threshold in the UKBB provided evidence of pathogenicity ...” As stated above, this and similar statements should be toned down.

I 245: “The large effect size threshold provided evidence for pathogenicity ...” See above.

Discussion

I 260: “A large variant effect size threshold provided evidence for pathogenicity (...) providing evidence for pathogenicity for variants not previously subjected to rigorous clinical assertion processes.” Again, please tone down.

I. 266: “First, quantitative endophenotypes (...) can be leveraged to infer rare variant pathogenicity.” See above.

I. 268: “We demonstrated that effect sizes (...) are proxies for variant pathogenicity” See above.

I. 295: The authors rightly indicate that further analyses would be required to compare the predictive utility of allelic effect size against other computational tools. It would be highly informative to have these comparisons in the context of this work.

Methods

I 374: When describing that HWE was tested with a specific threshold, please specify the used test and justify the used threshold.

I 378: Please give a reference for the KING algorithm. Also, please provide details about the parameter settings instead of only describing that this algorithm was used.

I 383: Please describe why outliers were defined by lying outside of 8 standard deviations from the mean.

I 396: When describing the imputation of missing values it needs to be specified for how many participants values were imputed. Also, when giving the baseline characteristics in table 1 and the supplementary tables, specify whether these are already based on imputed values.

I 439: Given that only rare variants were analysed, it is recommended not to use an additive coding, because the homozygous genotypes will be extremely rare, leading to unstable estimates. Coding carriers versus non-carriers in a dominant model would be preferred.

I 472: Were the effects sizes approximately normally distributed to allow for an ANOVA? In general, the rationale for these analyses was not clear to me if followed up by the regression analyses.

Figures

Figure 1 (Schema of study design). This figure is not informative and could be removed entirely.

Figure 2, upper row: How were the curves estimated or smoothed for this illustration? For this figure, box plots of the distributions are recommended instead.

Figure 2, lower row: Here, box plots of the estimated values showing mean, median, relevant quantiles and extremes are recommended, given that these would better illustrate the range of values. In fact, if the differences to B are shown, what does the height of the box of B indicate? And how can the box indicate the difference, since the box shows a range instead of a single value? Why do not all of the boxes have confidence intervals/whiskers? Also, instead of indicating p-values below 5% with an asterisk, please state the exact p-values.

Figure 3: Please include information on the numbers of the variants.

Reviewer #2 (Remarks to the Author):

This study uses large biobank cohorts of genotyped and phenotyped individuals to address an intriguing question - for diseases with disease relevant endophenotypes, can such measures be used to inform pathogenicity of rare variants and improve variant classification? This is potentially a very beneficial use of biobank data to address a critical challenge in genomic medicine, namely the VUS crisis. Three important and relevant diseases are used to validate this approach and the authors demonstrate that clear effects on the endophenotypes can be observed for known disease-causing variants (and nicely control this with a panel of cancer genes). They then identify a large number of variants (both novel variants and those with uncertain classification) with apparent large effects on the endophenotypes and suggest this evidence can be used to inform the possible pathogenicity of these variants.

I have a number of concerns with the approaches and methods used in this study however and with the utility of this approach in contributing to variant interpretation.

The authors use a very lenient frequency threshold (MAF of 1%) to define rare variants. It is likely that most disease relevant variants in the analysed genes will be much rarer than this threshold. The main analysis (Figure 2) highlights the effect of this approach, i.e. huge numbers of carriers with B/LB/VUS/C variants, most of which are likely to be quite common (between 0.1% and 1%), have little to no effect on phenotype and therefore help to drive the clear distinction in phenotypes between the P/LP variants and the other groups. The challenge in rare variant interpretation is not to distinguish pathogenic ultra-rare variants from benign non-rare or low frequency variants but rather to distinguish between pathogenic and benign variants that are equally ultra-rare. The authors do show a sensitivity analysis for variants with a MAC<6. However, this is quite a blunt approach (why this number?) and doesn't take into account the different characteristics and genetic architectures of the diseases in question (such as the different prevalences of FH and LQTS/MODY). And indeed the sensitivity analysis does show less clear differences between the P/LP and other groups. There are several variants in Supp Tables 10 and 11 which are observed in numerous individuals and are likely to be too common in the population to be penetrant pathogenic variants. While they may still contribute to disease with a more intermediate effect size, this limitation/issue should be noted.

Figure 2 demonstrates clear associations between endophenotypes and variants classified as P/LP. Given the known pathogenicity of these variants, this is an expected finding but one that helps to validate the approach of this study. I am surprised however by the lack of any intermediate association for the VUS or conflicting variants in this analysis, given that later in the manuscript the authors demonstrate that 10%-25% of these variants passed the large effect threshold and therefore provided evidence of pathogenicity. How do the authors explain this apparent discrepancy? Is it possibly related to the lenient frequency threshold (1%) used to define rare variants such that there are very large numbers of carriers of VUS/C variants (likely the more common ones) and any signal from the putatively pathogenic VUS/C is therefore diluted?

Endophenotype evidence for pathogenicity is demonstrated for 268 VUS across the 3 disease gene sets, the vast majority of which were missense. When assessing for variants not previously submitted to ClinVar, a further 843 variants had apparent large effect sizes. However, 27.8% of these variants were synonymous. While some synonymous variants can affect gene function (e.g. by affecting splicing), it is very surprising to see such a large proportion with an apparent effect and calls into question the specificity of this approach. How do the authors explain such a finding? Also, see minor points below for some concerns about the variant consequence classifications.

Replication cohorts - while broadly similar patterns were observed in the replications cohorts (pathogenic variants having significantly greater effect sizes than benign), it should be stated in the main text that no significant difference is observed for the LP variants in 2 of the 3 conditions analysed.

The authors argue that this approach could be used to inform pathogenicity but do not offer details as to how it could be implemented in practice. They rightly highlight the limitations of generic

computational predictors, but do not mention more sophisticated, disease-specific approaches that have been developed (e.g. case-control analysis, high throughput functional assays, machine learning techniques). Also, many of the VUS/C/novel variants that evidence of pathogenicity based on endophenotype data (Supp Tables 10/11) are observed in only 1 individual in the biobank datasets - how confident can we be in this effect on endophenotype if only seen in one person?

This study rightly focuses on genes with readily measurable and disease relevant phenotypes as a proof of concept for this approach. In the discussion, the authors state "We anticipate that our findings will be widely applicable beyond FH, LQTS and MODY to additional quantitative endophenotypes for other monogenic diseases". I wonder how widely applicable this approach will be to other diseases that are less likely to have endophenotypes as directly translatable as QTc and LDL-C levels. This doesn't negate using such an approach where endophenotypes are relevant, but if you're going to claim a widespread utility for the approach, this should be justified by explaining how this will be done for different diseases and what those endophenotypes are.

Minor points:

Introduction - the authors state that the ACMG guidelines are "laborious and prone to adjudicator disagreement" - while true, the major cause of the abundance of VUS is the genetic heterogeneity of diseases and the lack of variant specific evidence for pathogenicity given that many variants are observed in only a few (or even single) patients/families.

Expanded analysis to commercial panel genes - the results for this sensitivity analysis were similar to the main findings unexpectedly, as there are unlikely to be many P/LP variants in the minor disease genes and even fewer erroneous variant classifications in ClinVar. Nevertheless, the use of commercial panels as a proxy for a broader disease gene set is problematic - commercial panels are often designed to maximise size rather than based on robustness of the gene-disease associations. Results from ClinGen re-classifications would be more appropriate (if available for the diseases studied here). As stated above, this is unlikely to change any results so is of minor importance, but would be a more appropriate method.

Supp Table 11 - there appear to be some errors in the Trait column, e.g. LDL for several of the LQTS genes. Also, the consequences of the variants should be checked (I noticed several variants where the apparent consequences did not match those in gnomAD for the variants in question). It would be useful to include HGVS descriptions here (and help with error checking for these consequences). gnomAD PopMax frequencies would also be useful.

Methods - ClinVar variants: "For variants with an aggregate classification in ClinVar as "benign/likely benign" or "pathogenic/likely pathogenic", the most recent submission classification was used to disaggregate the two classes". I'm not convinced of this approach to disaggregate these classes. While it is possible that the most up-to-date entry in ClinVar represents the most accurate classification, there is a great deal of variability between different labs and in the quality and comprehensiveness of the entries and the evidence underlying the classifications. I would suggest that going with the classification used by the majority of labs is likely to be more accurate, and perhaps only using those entries with evidence summaries as a proxy for high quality entries.

Reviewer #3 (Remarks to the Author):

Morrill et al. considered whether quantitative endophenotypes can support rare variant (here defined as <1% MAF) classification, for familial hypercholesterolemia (FH), long QT syndrome (LQTS) and maturity-onset diabetes of the young (MODY). The authors have undertaken three main analyses:

1. They show that for these monogenic diseases, population scale rare variant studies for endophenotypes (LDL-C, QTc and HbA1c, respectively) can discriminate between (likely) pathogenic and (likely) benign variants located within core disease genes (FH: LDLR, APOB, PCSK9; LQTS: KCNQ1, KCNH2, SCN5A; MODY: GCK, HNF1A, HNF1B, HNF4A).
2. They suggest implementation of an effect size threshold (>0.5 SD of the trait distribution) can reclassify variants of uncertain significance, as categorised on ClinVar, for FH (15.8%), LQTS (24.6%) and MODY (10.4%).
3. They suggest the method they describe is suitable for classifying additional variants, detected through their analysis but not previously reported in ClinVar, as putatively pathogenic for FH (21.3%), LQTS (30.7%) and MODY (15.1%).

The manuscript is well written and statistical analysis appears to be appropriately performed. Variants of uncertain significance represent a huge challenge for the clinical community and tools to help resolve these issues are in demand.

I have a few major concerns that I would like to see addressed.

1. Lines 274 – 276: "We anticipate that our findings will be widely applicable beyond FH, LQTS and MODY to additional quantitative endophenotypes for other monogenic diseases."

Analysis undertaken in part 1 is not surprising given that the monogenic diseases selected are underpinned by extremes in the corresponding endophenotype. But for such an approach to have further utility it will be important for the authors to define the assumptions they have made when selecting these diseases/endophenotypes and how the genetic architectures of these examples might allow for such an approach to be pursued. Please qualify the statement in lines 274-276 by explaining how others might be able to implement the methodology and how appropriate endophenotypes for monogenic diseases can be established. For example, how would such an approach be used to resolve variants of uncertain significance in other relatively common monogenic diseases, such as hypertrophic cardiomyopathy, where a range of related endophenotypes could be adopted?

2. There are many in silico tools that have attempted to reclassify variants of uncertain significance. The current analysis proposes a >0.5 SD effect size threshold is sufficient to re-classify variant status, but this needs to be benchmarked against other available tools within the community (including, but not limited to: CADD, Eigen, M-CAP, REVEL, PrimateAI). Whilst ClinVar is a useful resource, for the approach described by Morrill et al. to gain clinical credibility, formal blinded comparison against contemporary ACMG-based classification methods should be conducted, rather than a reliance on accurate variant classification status via ClinVar.

3. The authors consider rare variants to be those with $<1\%$ MAF in disease causing genes. Based on this, the variant set considered for re-classification includes many variants that would not typically be considered via ACMG criteria and may explain why 1) these variants did not feature in ClinVar; and 2) 27.8% of variants were reported to be synonymous (i.e. a surprising result for disease-causing variants). Prior work has demonstrated that allele frequency is important when considering variant classification status and the ClinGen familial hypercholesterolemia Expert Panel suggest the ACMG PM2 rule requires a MAF of $<0.02\%$ for LDLR (<https://www.medrxiv.org/content/10.1101/2021.03.17.21252755v1>). As such, the current approach documented for variants that do not feature in ClinVar is not appropriate and should either be removed or critically revised to align with community standards.

We would like to thank the editors and reviewers for their comments, which we feel have strengthened the manuscript.

In addition to the response to comments below, we have included the following updates to the manuscript:

- A group of participants in the UK Biobank withdrew their consent from the study in August 2021 just prior to submission of this manuscript; this revised manuscript has excluded those participants resulting in small changes to cohort numbers

Reviewer 1 Comments:

In their manuscript, the authors describe a novel approach to utilize endophenotype effect sizes to classify variants as pathogenic. For this, they use data for endophenotypes for three monogenic diseases in the UKBB data as well as matching replication cohorts. Given the large proportion of variants with unresolved pathogenicity, the approach is very interesting and potentially useful. Based on the currently presented material, however, the value of the approach is not yet clear because some methods are not described in sufficient detail, and statistical analyses should be complemented.

First, in general, thresholds applied in quality control and data analysis need to be justified since, in most cases, no generally accepted standards exist, and examples for that are given below.

Author response: Thank you for raising this important point -- we have included justifications in response to the reviewer's specific questions in the Methods section (below).

Second, it is unclear how the authors handle the multiple testing situation.

Author response: Effect size inference was not conditioned on statistical association given the rarity of variants included in our analysis.

Third, whenever possible, confidence intervals should be provided in addition to effect estimates. Specifically, providing confidence intervals for the estimates of the AUC, sensitivity and specificity should be given to facilitate the evaluation of the estimated values. For example, for LDL-C, the chosen threshold leads to a sensitivity of 75%, which is associated with a 95% confidence interval of about 60% to 87% in this sample size, and it then needs to be discussed whether these values are practically relevant.

Fourth, diagnostic values should be complemented by positive and negative predictive values to estimate the probability for a variant to be pathogenic given the effect size exceeds the given threshold. For the example of LDL-C, the PPV is roughly estimated to be below 40% if the a priori probability is taken from the proportion of pathogenic variants in all variants excluding those classified as VUS or C. The interpretation and the discussion of the results then needs to take these values into account.

Author response: We have included confidence intervals, sensitivity, specificity, PPV, and NPV, and interpretations of these values when appropriate in the manuscript.

Revised manuscript:

- Results:
 - "For the LDL-C endophenotype, variant effect sizes among FH genes discriminated pathogenic variants from variants classified as likely benign and benign in a logistic regression model with an AUC of 0.84 (95% CI 0.74, 0.93; 45 variants included) and 0.91 (95% CI 0.87, 0.96; 58 variants included) in the UKBB and FOURIER, respectively (Figure 2A). For the QTc interval, variant effect sizes among LQTS genes discriminated pathogenic variants with an AUC of 0.83 (95% CI 0.71, 0.95; 20 variants included) and 0.79 (95% CI 0.66, 0.92; 23

variants included) in the UKBB and TOPMed, respectively (Figure 2B). For HbA1c, variant effect sizes among MODY genes discriminated pathogenic variants with an AUC of 0.82 (95% CI 0.67, 0.97; 20 variants included) and 0.91 (95% CI 0.76, 1; 2 variants included) in the UKBB and FOURIER, respectively (Figure 2C)."

- "In the UKBB, the large LDL-C effect size threshold had a sensitivity of 76% (95% CI 63, 88) and specificity of 88% (95% CI 85, 91) for discriminating ClinVar designated pathogenic variants from non-pathogenic variants (likely benign, and benign). PPV and NPV were calculated to be 37% (95% CI 27, 47) and 97% (95% CI 96, 99) respectively."
- "The large effect size threshold corresponding to 0.5 SD of the QTc distribution in the UKBB was 11.9 ms, which had a sensitivity of 80% (95% CI 62, 98) and specificity of 74% (95% CI 69, 79), with PPV of 19% (95% CI 10, 27) and NPV of 98% (95% CI 96, 100)."
- "For HbA1c, the large effect threshold was 0.31%, with a sensitivity of 70% (95% CI 50, 90), specificity of 95% (95% CI 89, 100), PPV of 82% (95% CI 64, 100), and NPV 90% (95% CI 82, 98)..."
- Discussion: "We acknowledge that effect size thresholds may differ by disease, given test characteristics that vary according to effect size threshold and endophenotype. Further analyses are warranted to examine the prognostic implications of large-effect variation, test different effect size thresholds for screening potential pathogenicity, and discover easily ascertainable endophenotypes for other monogenic diseases to aid in variant classification."
- Supplemental Table 9 now includes confidence intervals for sensitivity, specificity, PPV, and NPV

Fifth, much of the interpretation of the results focuses on the proportion of as yet unresolved variants to be classified as pathogenic given the defined thresholds. It should be made clear why this is an important result. The proportion of variants classified as pathogenic could easily be varied by using other thresholds, and the thresholds used are not optimized with regard to classification accuracy. Also, it is unclear how the algorithm including the defined thresholds would perform in currently unresolved variants, given that these are likely to systematically differ from those variants for which the pathogenicity is known.

Author response: We acknowledge that the proportion of variants screened as potentially pathogenic varies with different effect size thresholds and had chosen the 0.5 SD threshold *a priori* for the primary analysis. We then conducted a sensitivity analysis exploring the sensitivity, specificity, PPV, and NPV of different SD thresholds to demonstrate the range of classification accuracy observed. The goal of this analysis was to demonstrate application of the 0.5 SD acknowledging that this may not be the "optimal" threshold for every endophenotype examined or in every context. We have included more discussion regarding this result. We acknowledge that effect size thresholds may be best selected in conjunction with other tools such as prediction algorithms, minor allele frequencies, etc., to screen variants not previously submitted to ClinVar for potential pathogenicity.

Relevant manuscript text:

- Results: "A *priori*, we selected variant effect size thresholds that corresponded to 0.5 SD of the endophenotype distribution in the UKBB. As a sensitivity analysis, we tabulated the sensitivity, specificity, positive predictive value (PPV), and negative predictive value (NPV) of other effect size thresholds based on a range of SD thresholds (Supplemental Table 9)."

Revised manuscript:

- Discussion:
 - “We anticipate that effect size may be utilized either prior to initiating a formal variant classification process or in conjunction with existing methods, all of which will require further prospective evaluation.”
 - “We acknowledge that effect size thresholds may differ by disease, given test characteristics that vary according to effect size threshold and endophenotype. Further analyses are warranted to examine the prognostic implications of large-effect variation, test different effect size thresholds for screening potential pathogenicity, and discover easily ascertainable endophenotypes for other monogenic diseases to aid in variant classification.”
- Supplemental Table 9 now includes confidence intervals for sensitivity, specificity, PPV, and NPV

Finally, many statements in the results interpretation and discussion need to be toned down, and examples for that are given below.

Introduction

I 102: It is stated that “quantitative endophenotypes are inherently more powerful phenotypes for genetic association than dichotomous disease status indicators”. In general, it is true that the analysis of quantitative traits is more powerful than the analysis of qualitative traits. However, this cannot directly be translated to endophenotypes versus binary disease states, because this additionally depends on the strength of the association between the endophenotype and the disease state.

Author response: We agree with the reviewer and have modified the statement below.

Revised manuscript:

- Introduction: “Quantitative endophenotypes are powerful phenotypes for genetic association and are often more easily and reliably ascertained than dichotomous disease status indicators.”

I 111: Please discuss whether the selected monogenic diseases are representative for others; does the selection of those diseases for which easily ascertained human-derived endophenotypes exist restrict the generalizability of the results?

Author response: We have removed the statement “We anticipate that our findings will be widely applicable beyond FH, LQTS and MODY to additional quantitative endophenotypes for other monogenic diseases” and added more nuanced justification to the Discussion.

Revised manuscript:

- Discussion: “The three diseases studied have precisely defined endophenotypes which are heritable, associated with the mechanism of disease, and included in the diagnostic criteria for the syndrome. We anticipate that our findings will be most effectively applied for diseases with readily measurable endophenotypes in which the endophenotype is highly correlated with disease status, including other cardiac diseases such as aortic aneurysm and various cardiomyopathies.^{1,2} We hypothesize that the application of endophenotype effect sizes to inferring pathogenicity may also be relevant for diseases in which endophenotypes complement diagnostic criteria for diseases.³ Future studies are required to characterize the potential applications of the described approach and define criteria for clinically informative endophenotypes.”

Results

I 226: “The large effect variant threshold in the UKBB provided evidence of pathogenicity ...” As stated above, this and similar statements should be toned down.

I 245: “The large effect size threshold provided evidence for pathogenicity ...” See above.

Discussion

I 260: “A large variant effect size threshold provided evidence for pathogenicity (...) providing evidence for pathogenicity for variants not previously subjected to rigorous clinical assertion processes.” Again, please tone down.

I. 266: “First, quantitative endophenotypes (...) can be leveraged to infer rare variant pathogenicity.” See above.

I. 268: “We demonstrated that effect sizes (...) are proxies for variant pathogenicity” See above.

Author response: We thank the reviewer for their perspective and have toned down such statements.

Revised manuscript:

- Abstract: “An effect size threshold of ≥ 0.5 times the endophenotype standard deviation nominated up to 35% of rare variants of uncertain significance or not in ClinVar in disease susceptibility genes with pathogenic potential.”
- Results:
 - “...we applied the large variant effect size threshold to novel variants that were not previously submitted to ClinVar as a screen for potential pathogenicity.”
- Discussion:
 - “Additionally, up to 30% of variants without ClinVar assertions had large effect sizes, providing potential for pathogenicity for variants not previously subjected to rigorous clinical assertion processes.”
 - “We demonstrated that effect sizes for rare variants in FH, LQTS, and MODY monogenic disease susceptibility genes with corresponding endophenotypes are associated with variant pathogenicity in three large studies.”
 - “Further, variant effect sizes may be useful in screening for potential pathogenicity of novel variants.”
 - “In conclusion, population-based genetic association testing for monogenic disease endophenotypes may enable scalable inferences that provide evidence toward variant pathogenicity. Future analyses are warranted to test whether large effect size variants are associated with clinical outcomes, and whether variant effect size information can be implemented in variant pathogenicity assertion workflows.”

I. 295: The authors rightly indicate that further analyses would be required to compare the predictive utility of allelic effect size against other computational tools. It would be highly informative to have these comparisons in the context of this work.

Author response: We thank the reviewer for this important comment and have added additional analyses characterizing the functional impact of the reclassified missense variants of uncertain significance or conflicting assertions using a suite of *in silico* tools from the dbNSFP database. We are planning future studies to assess the impact of adding effect size as a tool to increase accuracy and/or efficiency of contemporary ACMG-based classification methods.

Revised manuscript:

- Methods: “We characterized the potential functional consequences of the non-synonymous variants with pathogenic potential by aggregating the output of 31 *in silico* prediction algorithms in dbNSFP v4.2a.⁴ into a predicted functional impact (PFI) score. Both qualitative (SIFT, SIFT4G, Polyphen2 HDIV, Polyphen2 HVAR, LRT,

MutationTaster, FATHMM, PROVEAN, MetaSVM, MetaLR, MetaRNN, M-CAP, PrimateAI, deogen2, BayesDel addAF, BayesDel noAF, ClinPred, LIST-S2, fathmm-MKL, fathmm-XF, MutationAssessor, and ALoFT) and quantitative (VEST v4.0, REVEL, MutPred, MVP, MPC, DANN, CADD, Eigen, and Eigen-PC) prediction algorithms were included. Each missense variant gained 1 point per algorithm if predicted to have a functional impact (designated as “D” for qualitative tools, “H” for MutationAssessor, and > 90% for quantitative tools); additional details regarding functional impact scores and cutoffs are publicly available.⁴ When algorithms did not generate a prediction for the variant of interest, they received an “NA” designation. PFI scores were then calculated for each variant by dividing the number of bioinformatic tools predicting the variant to have a functional consequence by the total number of bioinformatic tools with variant prediction information available such that scores ranged from 0-1, with higher values indicating greater predicted impact. Heatmaps were generated to display the *in silico* predictions of functional consequences of VUS and conflicting variants using ggplot2 in R version 3.6.”

- Results:

- “In aggregate, of the 253 VUS across FH, LQTS, and MODY genes for which large effect size thresholds provided evidence of pathogenicity, there were 244 (96.4%) missense variants, 7 (2.8%) synonymous variants, 1 (0.4%) in-frame deletion, and 1 (0.4%) in-frame insertion. We created an aggregate measure of 31 bioinformatic tools designed to predict function, which we denoted as the predicted functional impact (PFI) score which ranged from 0-1; higher PFI scores indicate greater predicted impact (see Methods for full detail). Of the 119 missense variants in genes associated with FH, the median PFI was 0.28 (Q1-Q3 0.09-0.41). The highest aggregate PFI was observed for variants in LDLR (median PFI 0.58, Q1-Q3 0.40-0.71). Of the 114 missense variants in genes associated with LQTS, the median PFI was 0.49 (Q1-Q3 0.30-0.69). Of the 11 missense variants in genes associated with MODY, the median PFI was 0.54 (Q1-Q3 0.44-0.81), with variants in HNF1B displaying the highest PFI of around 0.82. Detailed bioinformatic tool functional predictions are shown in Supplemental Figure 7; additional variant characteristics are listed in Supplemental Table 10.”
- “There were 139 non-synonymous variants with conflicting assertions in ClinVar across FH, LQTS, and MODY genes for which large effect size thresholds provided evidence of pathogenicity. Of these, 112 (80.6%) were missense variants, 22 (15.8%) were synonymous variants, 2 (1.4%) were splice donor variants, 2 were frameshift or stop-gained variants (1.4%), and 1 (0.7%) was an in-frame deletion. There were 71 missense variants in genes associated with FH with median PFI 0.63 (Q1-Q3 0.25-0.77), with the highest PFI observed again in LDLR variants (n=53, median PFI 0.76, Q1-Q3 0.57-0.80). There were 38 missense variants in genes associated with LQTS with median PFI 0.66 (Q1-Q3 0.33-0.82) and 4 missense variants in genes associated with MODY with median PFI 0.80 (Q1-Q3 0.66-0.86). Detailed bioinformatic tool functional predictions are shown in Supplemental Figure 8; additional variant characteristics including minor allele count are listed in Supplemental Table 10.”
- “Lastly, we applied the large variant effect size threshold to novel variants that were not previously submitted to ClinVar as a screen for potential pathogenicity. In the UKBB, 438 (21.3%), 99 (30.4%), and 124 (14.3%) variants not previously submitted to ClinVar were found to have large effect size in genes associated with FH, LQTS, and MODY, respectively. Similar percentages were observed in

the replication datasets (18.8%, 19.4%, and 20.1%, respectively). In total 807 large effect size variants were identified, of which 544 (67.4%) were missense, 237 (29.4%) synonymous, 22 (2.7%) were loss-of-function (LOF; frameshift or stop-gained), and the remainder with various consequences. For the 355 non-synonymous variants in FH genes, 339 (95.5%) were missense and 14 (3.9%) were LOF, with a median PFI of 0.28 (Q1-Q3 0.12-0.43). For the 106 non-synonymous variants in LQTS genes, there were 102 (96.2%) missense and 3 LOF (2.8%), with median PFI of 0.56 (Q1-Q3 0.41-0.73). There were 109 non-synonymous variants in the MODY genes, with 103 (94.5%) missense and 5 (4.6%) LOF, and median PFI of 0.63 (Q1-Q3 0.48-0.79). Additional details for the non-synonymous variants, including minor allele count, are provided in Supplemental Table 11.”

- Discussion: “While there is a growing landscape of computational tools being deployed to predict pathogenicity through assessment of variant conservation and prediction of variant effects on protein function, these tools remain imperfect and are not informed by features specific to a given disease.^{5,6} We observed that the PFI, an aggregate measure of predicted variant function, was variable and heterogeneous in relation to effect size.”
- Figures:

Supplemental Figure 7. Heatmap of bioinformatic tool functional predictions for variants of uncertain significance with large effect sizes

Heatmaps display functional predictions by bioinformatic tools for large effect-size variants of uncertain significance (VUS). Red boxes represent high predicted functional impact, yellow boxes represent low predicted functional impact, and grey boxes represent no prediction available. Panels A, B, and C display data for rare variants associated with LDL-C, QTc, and HbA1c endophenotypes, respectively. Definitive familial hypercholesterolemia (FH) genes include *LDLR*, *APOB*, *PCSK9*. Definitive long-QT syndrome (LQTS) genes include *KCNQ1*, *KCNH2*, *SCN5A*. Common maturity-onset diabetes of the young (MODY) genes include *HNF1A*, *HNF1B*, *HNF4A*, *GCK*; the color bar indicates the corresponding gene for the variants shown.

Supplemental Figure 8. Heatmap of bioinformatic tool functional predictions for variants with conflicting assertions with large effect sizes

Heatmaps display functional predictions by bioinformatic tools for large effect-size variants with conflicting assertions in ClinVar. Red boxes represent high predicted functional impact, yellow boxes represent low predicted functional impact, and grey boxes represent no prediction available. Panels A, B, and C display data for rare variants associated with LDL-C, QTc, and HbA1c endophenotypes, respectively. Definitive familial hypercholesterolemia (FH) genes include *LDLR*, *APOB*, *PCSK9*. Definitive long-QT syndrome (LQTS) genes include *KCNQ1*, *KCNH2*, *SCN5A*. Common maturity-onset diabetes of the young (MODY) genes include *HNF1A*, *HNF1B*, *HNF4A*, *GCK*; the color bar indicates the corresponding gene for the variants shown.

Methods

I 374: When describing that HWE was tested with a specific threshold, please specify the used test and justify the used threshold.

I 378: Please give a reference for the KING algorithm. Also, please provide details about the parameter settings instead of only describing that this algorithm was used.

I 383: Please describe why outliers were defined by lying outside of 8 standard deviations from the mean.

Author response: We have included the reference and parameter for the KING algorithm. The HWE threshold and outlier definition were used in prior studies using UKBB exomes which are cited below.

Revised manuscript:

- Methods:
 - “UKBB variants were removed if they were in low complexity regions, had call rates <90%, failed the Hardy Weinberg Equilibrium test ($P \leq 1.0 \times 10^{-15}$), or were monomorphic in the final dataset, as previously described.^{7,8}”
 - “Duplicate individuals in the UKBB were identified with KING⁹ (-- duplicate) and removed if not a monozygotic twin. Genetically determined sex was calculated using high quality (MAF $\geq 0.1\%$, missingness $\leq 1\%$, Hardy Weinberg Equilibrium $P \geq 10^{-6}$) independent variants on the X chromosome, as described in detail in previous studies.^{7,8,10,11}
 - “Samples were removed if they were outliers (outside of 8 standard deviations (SD) from the mean)¹⁰ for quality control metrics including heterozygosity homozygosity ratio, transition and transversion ratio, SNP and Indel ratio, and the number of singletons per sample.”

I 396: When describing the imputation of missing values it needs to be specified for how many participants values were imputed. Also, when giving the baseline characteristics in table 1 and the supplementary tables, specify whether these are already based on imputed values.

Author response: We have revised the manuscript to include the number of imputed values in the CONSORT diagrams (Supplemental Figures 1-3, see below) and specified in the baseline cohort characteristic tables (Table 1, Supplemental Table 1) that the reported data included median imputed values.

Relevant manuscript text:

- Methods:
 - Clinical trait exclusions and covariates
 - “For our LDL-C analysis...[w]e imputed participants with incomplete data for HDL to the median HDL value. We imputed participants with incomplete data for history of myocardial infarction and history of statin usage to no history.”
 - “For our QTc analysis...[w]e imputed participants with incomplete data for history of myocardial infarction and history of heart failure to no history.”
 - “For our HbA1c analysis...[w]e imputed participants with incomplete data for MCV to the median MCV value.”

Revised manuscript:

- Figures:
 - Supplemental Figure 1.** CONSORT diagram of LDL-C endophenotype analysis

Supplemental Figure 2. CONSORT diagram of QTc endophenotype analysis

Supplemental Figure 3. CONSORT diagram of HbA1c endophenotype analysis

I 439: Given that only rare variants were analysed, it is recommended not to use an additive coding, because the homozygous genotypes will be extremely rare, leading to unstable estimates. Coding carriers versus non-carriers in a dominant model would be preferred.

Author response: We used an additive model for our primary analysis but also tested a dominant model for each trait; the Pearson coefficient for effect size was 1 for all three traits. We have included plots below showing the distribution of variant effect size for each testing model and corresponding correlation plot. Given the near-perfect correlation of effect size, we chose to use the additive genetic model.

I 472: Were the effect sizes approximately normally distributed to allow for an ANOVA? In general, the rationale for these analyses was not clear to me if followed up by the regression analyses.

Author response: Thank you for raising this important point. We have removed ANOVA testing from the manuscript and the associated figures.

Figures

Figure 1 (Schema of study design). This figure is not informative and could be removed entirely.

Author response: We have removed this figure accordingly

Figure 2, upper row: How were the curves estimated or smoothed for this illustration? For this figure, box plots of the distributions are recommended instead.

Author response: These are kernel density curves generated using ggplot2's geom_line() function (https://www.rdocumentation.org/packages/ggplot2/versions/0.9.0/topics/geom_line) with statistical transformation set to density (stat = "density"). We used the default smoothing parameter (kernel bandwidth = 1) and confirmed appropriate smoothing by comparing histogram distributions against the density curve (see below for example of density curve compared to histogram for variants in definitive FH genes). We chose to display the effect size distributions using density curves instead of a histogram for illustrative clarity.

Figure 2, lower row: Here, box plots of the estimated values showing mean, median, relevant quantiles and extremes are recommended, given that these would better illustrate the range of values. In fact, if the differences to B are shown, what does the height of the box of B indicate? And how can the box indicate the difference, since the box shows a range instead of a single value? Why do not all of the boxes have confidence intervals/whiskers? Also, instead of indicating p-values below 5% with an asterisk, please state the exact p-values.

Author response: We would like to clarify that these plots show results from linear regression models in which we regressed the individual endophenotype value against the carrier status of each individual for variants in each pathogenicity category, using Benign variants as the referent group. As such, the plot shows the estimated difference in individual endophenotype level for carriers of variants of different classifications, compared to carriers of Benign variants, the referent group). We recognize that using a box plot was not our initial intention, and have changed this to a dot plot displaying the mean adjusted endophenotype effect size and 95% confidence interval. Some of the confidence intervals are narrow and thus covered by the point estimate marker; we have decreased the size of the marker so that the confidence intervals can be appreciated. We annotated exact p-values for the categories with p-value < 0.05. Lastly, we edited the figure captions to reflect these changes.

Relevant manuscript text:

- Methods: "We next assessed the estimated difference in individual endophenotype value for carriers of variants in each pathogenicity category compared to carriers of benign variants using a linear regression model. In the model, we regressed individual endophenotype values against carrier status of variants in each pathogenicity category, adjusting for age, sex, clinical covariates, and first 12 principal components of ancestry."

Revised manuscript:

- Methods: we have clarified that carriers of Benign variants served as the referent group: "...our referent group was designated as carriers of Benign (B) variants."
- Figures:

Figure 3: Please include information on the numbers of the variants.

Author response: we have included the number of variants on the revised figure.

Revised manuscript:

Reviewer 2 Comments:

This study uses large biobank cohorts of genotyped and phenotyped individuals to address an intriguing question - for diseases with disease relevant endophenotypes, can such measures be used to inform pathogenicity of rare variants and improve variant classification? This is potentially a very beneficial use of biobank data to address a critical challenge in genomic medicine, namely the VUS crisis. Three important and relevant diseases are used to validate this approach and the authors demonstrate that clear effects on the endophenotypes can be observed for known disease-causing variants (and nicely control this with a panel of cancer genes). They then identify a large number of variants (both novel variants and those with uncertain classification) with apparent large effects on the endophenotypes and suggest this evidence can be used to inform the possible pathogenicity of these variants.

I have a number of concerns with the approaches and methods used in this study however and with the utility of this approach in contributing to variant interpretation.

The authors use a very lenient frequency threshold (MAF of 1%) to define rare variants. It is likely that most disease relevant variants in the analysed genes will be much rarer than this threshold. The main analysis (Figure 2) highlights the effect of this approach, i.e. huge numbers of carriers with B/LB/VUS/C variants, most of which are likely to be quite common (between 0.1% and 1%), have little to no effect on phenotype and therefore help to drive the clear distinction in phenotypes between the P/LP variants and the other groups. The challenge in rare variant interpretation is not to distinguish pathogenic ultra-rare variants from benign non-rare or low frequency variants but rather to distinguish between pathogenic and benign variants that are equally ultra-rare. The authors do show a sensitivity analysis for variants with a $MAC < 6$. However, this is quite a blunt approach (why this number?) and doesn't take into account the different characteristics and genetic architectures of the diseases in question (such as the different prevalences of FH and LQTS/MODY). And indeed the sensitivity analysis does show less clear differences between the P/LP and other groups. There are several variants in Supp Tables 10 and 11 which are observed in numerous individuals and are likely to be too common in the population to be penetrant pathogenic variants. While they may still contribute to disease with a more intermediate effect size, this limitation/issue should be noted.

Author response: We have decreased the rare variant frequency threshold to $MAF < 0.1\%$. We have removed the sensitivity analysis for variants with $MAC < 6$. The net effect is that the number of P/LP variants have remained the same, while numbers of B/LB/VUS/C variants have decreased. Our findings with regard to carriers of P/LP variants and P/LP variant effect size compared to other variants remain consistent.

Figure 2 demonstrates clear associations between endophenotypes and variants classified as P/LP. Given the known pathogenicity of these variants, this is an expected finding but one that helps to validate the approach of this study. I am surprised however by the lack of any intermediate association for the VUS or conflicting variants in this analysis, given that later in the manuscript the authors demonstrate that 10%-25% of these variants passed the large effect threshold and therefore provided evidence of pathogenicity. How do the authors explain this apparent discrepancy? Is it possibly related to the lenient frequency threshold (1%) used to define rare variants such that there are very large numbers of carriers of VUS/C variants (likely the more common ones) and any signal from the putatively pathogenic VUS/C is therefore diluted?

Author response: While there is a small fraction of VUS and conflicting variants which passed the large effect size threshold, there are many more of such variants with negative and/or low effect size which dilute the signal. This pattern remains using a more stringent MAF threshold of $< 0.1\%$.

Endophenotype evidence for pathogenicity is demonstrated for 268 VUS across the 3 disease gene sets, the vast majority of which were missense. When assessing for variants not previously submitted to ClinVar, a further 843 variants had apparent large effect sizes. However, 27.8% of these variants were synonymous. While some synonymous variants can affect gene function (e.g. by affecting splicing), it is very surprising to see such a large proportion with an apparent effect and calls into question the specificity of this approach. How do the authors explain such a finding? Also, see minor points below for some concerns about the variant consequence classifications.

Author response: The proportion of synonymous variants with large effect sizes that were not previously submitted to ClinVar was similar after decreasing the MAF filter to < 0.1% (29.4%), as compared to a MAF filter of < 1% (29.0%). The 29.0% is different from the reviewer's cited 27.8% as we have uncovered an error in our code for variant consequence annotation which has now been fixed. 3.8% of synonymous variants with large effect sizes are located in splice regions, compared to 2.2% of synonymous variants with small effect size (Fisher's exact test p-value = 0.16). Nevertheless, we cannot exclude that synonymous variants with large effect sizes are more likely to confer a true biological impact than synonymous variants without large effect sizes. We also acknowledge that rare coding variants may be subject to imprecise effect size estimation, which we now state in the discussion. Lastly, we posit that effect size may be best applied in conjunction with other tools such as prediction algorithms, minor allele frequencies, etc., to screen variants not previously submitted to ClinVar for potential pathogenicity.

Relevant manuscript text:

- Results: "In total 807 large effect size variants were identified, of which 544 (67.4%) were missense, 237 (29.4%) synonymous, 22 (2.7%) were loss-of-function (LOF; frameshift or stop-gained), and the remainder with various consequences."

Revised manuscript:

- Discussion:
 - "We anticipate that a potential application may be to utilize effect size information, when available, either prior to initiating a formal variant classification process or in conjunction with existing methods. Such applications will require prospective evaluation."
 - "We acknowledge that effect size thresholds may differ by disease, given test characteristics that vary according to effect size threshold and endophenotype. Further analyses are warranted to examine the prognostic implications of large-effect variation, test different effect size thresholds for screening potential pathogenicity, and discover easily ascertainable endophenotypes for other monogenic diseases to aid in variant classification."
 - "Rare coding variants may be subject to imprecise effect size estimation which we anticipate will improve with larger repositories of sequence and phenotypic data over time."

Replication cohorts - while broadly similar patterns were observed in the replications cohorts (pathogenic variants having significantly greater effect sizes than benign), it should be stated in the main text that no significant difference is observed for the LP variants in 2 of the 3 conditions analysed.

Author response: We have added relevant statements regarding the LP variants in the main text.

Revised manuscript:

- Results: "We observed similar relations between rare variant effect sizes for each endophenotype and variant pathogenicity categories (Supplemental Tables 6-8, Supplemental Figure 6). Carriers of pathogenic variants had significantly greater

endophenotype values compared to benign variant carriers for all endophenotypes. Carriers of likely pathogenic variants had significantly greater endophenotype values compared to benign variant carriers for LDL-C; no significant difference was observed for QTc or HbA1c.”

The authors argue that this approach could be used to inform pathogenicity but do not offer details as to how it could be implemented in practice. They rightly highlight the limitations of generic computational predictors, but do not mention more sophisticated, disease-specific approaches that have been developed (e.g. case-control analysis, high throughput functional assays, machine learning techniques). Also, many of the VUS/C/novel variants that evidence of pathogenicity based on endophenotype data (Supp Tables 10/11) are observed in only 1 individual in the biobank datasets - how confident can we be in this effect on endophenotype if only seen in one person?

Author response: We are encouraged by the development of more sophisticated and high-throughput approaches to predicting variant pathogenicity and look forward to seeing how these tools can be applied in clinical settings. In future work we are planning to assess the joint impact of effect size and conventional ACMG-based classification methods for classifying variants. We acknowledge that our power is limited by the rarity of variants of interest and have added this to our study’s limitations. We posit that effect size may be best applied in conjunction with other tools such as prediction algorithms, minor allele frequencies, etc., to screen variants and have provided minor allele count information for VUS, conflicting, and novel variants with evidence of pathogenicity in Supplemental Tables 10-11.

Relevant manuscript text:

- Discussion:
 - “We anticipate that a potential application may be to utilize effect size information, when available, either prior to initiating a formal variant classification process or in conjunction with existing methods. Such applications will require prospective evaluation.”
 - “To facilitate use of our results into potential clinical practice and research, we will make variant level association results from the present analysis publicly accessible in the Cardiovascular Disease Knowledge Portal.¹² We expect that as the number of sequenced individuals grows, large compendia of variant effect sizes with endophenotypes may help classify variants as potentially pathogenic or benign, facilitating both research and clinical variant classification.”
- Supplement: Supplemental Table 10 and 11

Revised manuscript:

- Discussion:
 - “Rare coding variants may be subject to imprecise effect size estimation which we anticipate will improve with larger repositories of sequence and phenotypic data over time.”

This study rightly focuses on genes with readily measurable and disease relevant phenotypes as a proof of concept for this approach. In the discussion, the authors state "We anticipate that our findings will be widely applicable beyond FH, LQTS and MODY to additional quantitative endophenotypes for other monogenic diseases". I wonder how widely applicable this approach will be to other diseases that are less likely to have endophenotypes as directly translatable as QTc and LDL-C levels. This doesn't negate using such an approach where endophenotypes are relevant, but if you're going to claim a widespread utility for the approach, this should be justified by explaining how this will be done for different diseases and what those endophenotypes are.

Author response: We have removed the statement "We anticipate that our findings will be widely applicable beyond FH, LQTS and MODY to additional quantitative endophenotypes for other monogenic diseases" and added more nuanced justification to the Discussion.

Revised manuscript:

- Discussion: "The three diseases studied have precisely defined endophenotypes which are heritable, associated with the mechanism of disease, and included in the diagnostic criteria for the syndrome. We anticipate that our findings will be most effectively applied for diseases with readily measurable endophenotypes in which the endophenotype is highly correlated with disease status, including other cardiac diseases such as aortic aneurysm and various cardiomyopathies.^{1,2} We hypothesize that the application of endophenotype effect sizes to inferring pathogenicity may also be relevant for diseases in which endophenotypes complement diagnostic criteria.³ Future studies are required to characterize the potential applications of the described approach and define criteria for clinically informative endophenotypes."

Minor points:

Introduction - the authors state that the ACMG guidelines are "laborious and prone to adjudicator disagreement" - while true, the major cause of the abundance of VUS is the genetic heterogeneity of diseases and the lack of variant specific evidence for pathogenicity given that many variants are observed in only a few (or even single) patients/families.

Author response: We agree with the reviewer's points regarding the cause of increasing numbers of VUS variants and have included such reasons in the manuscript below.

Revised manuscript:

- Introduction: "In clinical practice, genetic testing is frequently hampered by discovery of VUS and conflicting variant classifications between laboratories, with lack of specific evidence for pathogenicity due to the genetic heterogeneity of disease and rarity of segregation data."

Expanded analysis to commercial panel genes - the results for this sensitivity analysis were similar to the main findings unexpectedly, as there are unlikely to be many P/LP variants in the minor disease genes and even fewer erroneous variant classifications in ClinVar. Nevertheless, the use of commercial panels as a proxy for a broader disease gene set is problematic - commercial panels are often designed to maximise size rather than based on robustness of the gene-disease associations. Results from ClinGen re-classifications would be more appropriate (if available for the diseases studied here). As stated above, this is unlikely to change any results so is of minor importance, but would be a more appropriate method.

Author response: the reviewer raises an important point regarding the limitations of using commercial panels as proxies for disease; we have removed this sensitivity analysis from the manuscript given these limitations and the lack of additional information provided by the results. ClinGen re-classification results are not available for all the monogenic diseases examined but we agree that this would be a more appropriate broader panel for future work.

Supp Table 11 - there appear to be some errors in the Trait column, e.g. LDL for several of the LQTS genes. Also, the consequences of the variants should be checked (I noticed several variants where the apparent consequences did not match those in gnomAD for the variants in question). It would be useful to include HGVS descriptions here (and help with error checking for these consequences). gnomAD PopMax frequencies would also be useful.

Author response: We thank the reviewer for their astute observation. We have uncovered an error in our code for variant consequence annotation which has now been fixed. We have also

included more variant characteristic descriptions including HGVS descriptions, gnomAD frequencies, and existing variant identifiers such as rsID.

Revised manuscript:

- Methods: "Variant consequences and relevant characteristics including HGVS descriptions and gnomAD minor allele frequencies were annotated using the Ensembl Variant Effect Predictor (VEP) tool.¹³"
- Supplement: Supplemental Table 11 has been updated

Methods - ClinVar variants: "For variants with an aggregate classification in ClinVar as "benign/likely benign" or "pathogenic/likely pathogenic", the most recent submission classification was used to disaggregate the two classes". I'm not convinced of this approach to disaggregate these classes. While it is possible that the most up-to-date entry in ClinVar represents the most accurate classification, there is a great deal of variability between different labs and in the quality and comprehensiveness of the entries and the evidence underlying the classifications. I would suggest that going with the classification used by the majority of labs is likely to be more accurate, and perhaps only using those entries with evidence summaries as a proxy for high quality entries.

Author response: We agree with the importance of including the highest quality variant assertion from ClinVar. We found that variants with aggregate assertions such as "benign/likely benign" or "pathogenic/likely pathogenic" were rare in ClinVar. In fact, none of the variants included in our analysis in the genes associated with FH, LQTS, or MODY had this aggregate assertion. As such, we have removed this line from the manuscript to decrease confusion.

Reviewer 3 Comments:

Morrill et al. considered whether quantitative endophenotypes can support rare variant (here defined as <1% MAF) classification, for familial hypercholesterolemia (FH), long QT syndrome (LQTS) and maturity-onset diabetes of the young (MODY). The authors have undertaken three main analyses:

- 1. They show that for these monogenic diseases, population scale rare variant studies for endophenotypes (LDL-C, QTc and HbA1c, respectively) can discriminate between (likely) pathogenic and (likely) benign variants located within core disease genes (FH: LDLR, APOB, PCSK9; LQTS: KCNQ1, KCNH2, SCN5A; MODY: GCK, HNF1A, HNF1B, HNF4A).*
- 2. They suggest implementation of an effect size threshold (>0.5 SD of the trait distribution) can reclassify variants of uncertain significance, as categorised on ClinVar, for FH (15.8%), LQTS (24.6%) and MODY (10.4%).*
- 3. They suggest the method they describe is suitable for classifying additional variants, detected through their analysis but not previously reported in ClinVar, as putatively pathogenic for FH (21.3%), LQTS (30.7%) and MODY (15.1%).*

The manuscript is well written and statistical analysis appears to be appropriately performed. Variants of uncertain significance represent a huge challenge for the clinical community and tools to help resolve these issues are in demand.

Author response: Thank you for your careful review of our manuscript and constructive comments.

I have a few major concerns that I would like to see addressed.

1.Lines 274 – 276: "We anticipate that our findings will be widely applicable beyond FH, LQTS and MODY to additional quantitative endophenotypes for other monogenic diseases."

Analysis undertaken in part 1 is not surprising given that the monogenic diseases selected are underpinned by extremes in the corresponding endophenotype. But for such an approach to have further utility it will be important for the authors to define the assumptions they have made when selecting these diseases/endophenotypes and how the genetic architectures of these examples might allow for such an approach to be pursued. Please qualify the statement in lines 274-276 by explaining how others might be able to implement the methodology and how appropriate endophenotypes for monogenic diseases can be established. For example, how would such an approach be used to resolve variants of uncertain significance in other relatively common monogenic diseases, such as hypertrophic cardiomyopathy, where a range of related endophenotypes could be adopted?

Author response: We agree with the reviewer's point that this method is best suited for diseases with diagnostic criteria that includes changes in an endophenotype compared to the general population, and that this method will require more testing for diseases with less precisely defined endophenotypes. We have removed the statement "We anticipate that our findings will be widely applicable beyond FH, LQTS and MODY to additional quantitative endophenotypes for other monogenic diseases" and added more nuanced justification to the Discussion.

Relevant manuscript text:

- Discussion: "Further analyses are warranted to examine the prognostic implications of large-effect variation, test different effect size thresholds for screening potential pathogenicity, and discover easily ascertainable endophenotypes for other monogenic diseases to aid in variant classification."

Revised manuscript:

- Discussion: "The three diseases studied have precisely defined endophenotypes which are heritable, associated with the mechanism of disease, and included in the diagnostic criteria for the syndrome. We anticipate that our findings will be most effectively applied for diseases with readily measurable endophenotypes in which the endophenotype is highly correlated with disease status, including other cardiac diseases such as aortic aneurysm and various cardiomyopathies.^{1,2} We hypothesize that the application of endophenotype effect sizes to inferring pathogenicity may also be relevant for diseases in which endophenotypes complement diagnostic criteria.³ Future studies are required to characterize the potential applications of the described approach and define criteria for clinically informative endophenotypes."

2. There are many in silico tools that have attempted to reclassify variants of uncertain significance. The current analysis proposes a >0.5 SD effect size threshold is sufficient to re-classify variant status, but this needs to be benchmarked against other available tools within the community (including, but not limited to: CADD, Eigen, M-CAP, REVEL, PrimateAI). Whilst ClinVar is a useful resource, for the approach described by Morrill et al. to gain clinical credibility, formal blinded comparison against contemporary ACMG-based classification methods should be conducted, rather than a reliance on accurate variant classification status via ClinVar.

Author response: We thank the reviewer for this important comment and have added additional analyses characterizing the functional impact of the reclassified missense variants of uncertain significance or conflicting assertions using a suite of *in silico* tools from the dbNSFP database. We are planning future studies to assess the impact of adding effect size as a tool to increase accuracy and/or efficiency of contemporary ACMG-based classification methods.

Revised manuscript:

- Methods: "We characterized the potential functional consequences of the non-synonymous variants with pathogenic potential by aggregating the output of 31 in silico prediction algorithms in dbNSFP v4.2a.⁴ into a predicted functional impact (PFI) score. Both qualitative (SIFT, SIFT4G, Polyphen2 HDIV, Polyphen2 HVAR, LRT,

MutationTaster, FATHMM, PROVEAN, MetaSVM, MetaLR, MetaRNN, M-CAP, PrimateAI, deogen2, BayesDel addAF, BayesDel noAF, ClinPred, LIST-S2, fathmm-MKL, fathmm-XF, MutationAssessor, and ALoFT) and quantitative (VEST v4.0, REVEL, MutPred, MVP, MPC, DANN, CADD, Eigen, and Eigen-PC) prediction algorithms were included. Each missense variant gained 1 point per algorithm if predicted to have a functional impact (designated as “D” for qualitative tools, “H” for MutationAssessor, and > 90% for quantitative tools); additional details regarding functional impact scores and cutoffs are publicly available.⁴ When algorithms did not generate a prediction for the variant of interest, they received an “NA” designation. PFI scores were then calculated for each variant by dividing the number of bioinformatic tools predicting the variant to have a functional consequence by the total number of bioinformatic tools with variant prediction information available such that scores ranged from 0-1, with higher values indicating greater predicted impact. Heatmaps were generated to display the *in silico* predictions of functional consequences of VUS and conflicting variants using ggplot2 in R version 3.6.”

- Results:

- “In aggregate, of the 253 VUS across FH, LQTS, and MODY genes for which large effect size thresholds provided evidence of pathogenicity, there were 244 (96.4%) missense variants, 7 (2.8%) synonymous variants, 1 (0.4%) in-frame deletion, and 1 (0.4%) in-frame insertion. We created an aggregate measure of 31 bioinformatic tools designed to predict function, which we denoted as the predicted functional impact (PFI) score which ranged from 0-1; higher PFI scores indicate greater predicted impact (see Methods for full detail). Of the 119 missense variants in genes associated with FH, the median PFI was 0.28 (Q1-Q3 0.09-0.41). The highest aggregate PFI was observed for variants in LDLR (median PFI 0.58, Q1-Q3 0.40-0.71). Of the 114 missense variants in genes associated with LQTS, the median PFI was 0.49 (Q1-Q3 0.30-0.69). Of the 11 missense variants in genes associated with MODY, the median PFI was 0.54 (Q1-Q3 0.44-0.81), with variants in HNF1B displaying the highest PFI of around 0.82. Detailed bioinformatic tool functional predictions are shown in Supplemental Figure 7; additional variant characteristics are listed in Supplemental Table 10.”
- “There were 139 non-synonymous variants with conflicting assertions in ClinVar across FH, LQTS, and MODY genes for which large effect size thresholds provided evidence of pathogenicity. Of these, 112 (80.6%) were missense variants, 22 (15.8%) were synonymous variants, 2 (1.4%) were splice donor variants, 2 were frameshift or stop-gained variants (1.4%), and 1 (0.7%) was an in-frame deletion. There were 71 missense variants in genes associated with FH with median PFI 0.63 (Q1-Q3 0.25-0.77), with the highest PFI observed again in LDLR variants (n=53, median PFI 0.76, Q1-Q3 0.57-0.80). There were 38 missense variants in genes associated with LQTS with median PFI 0.66 (Q1-Q3 0.33-0.82) and 4 missense variants in genes associated with MODY with median PFI 0.80 (Q1-Q3 0.66-0.86). Detailed bioinformatic tool functional predictions are shown in Supplemental Figure 8; additional variant characteristics including minor allele count are listed in Supplemental Table 10.”
- “Lastly, we applied the large variant effect size threshold to novel variants that were not previously submitted to ClinVar as a screen for potential pathogenicity. In the UKBB, 438 (21.3%), 99 (30.4%), and 124 (14.3%) variants not previously submitted to ClinVar were found to have large effect size in genes associated with FH, LQTS, and MODY, respectively. Similar percentages were observed in

the replication datasets (18.8%, 19.4%, and 20.1%, respectively). In total 807 large effect size variants were identified, of which 544 (67.4%) were missense, 237 (29.4%) synonymous, 22 (2.7%) were loss-of-function (LOF; frameshift or stop-gained), and the remainder with various consequences. For the 355 non-synonymous variants in FH genes, 339 (95.5%) were missense and 14 (3.9%) were LOF, with a median PFI of 0.28 (Q1-Q3 0.12-0.43). For the 106 non-synonymous variants in LQTS genes, there were 102 (96.2%) missense and 3 LOF (2.8%), with median PFI of 0.56 (Q1-Q3 0.41-0.73). There were 109 non-synonymous variants in the MODY genes, with 103 (94.5%) missense and 5 (4.6%) LOF, and median PFI of 0.63 (Q1-Q3 0.48-0.79). Additional details for the non-synonymous variants, including minor allele count, are provided in Supplemental Table 11.”

- Discussion: “While there is a growing landscape of computational tools being deployed to predict pathogenicity through assessment of variant conservation and prediction of variant effects on protein function, these tools remain imperfect and are not informed by features specific to a given disease.^{5,6} We observed that the PFI, an aggregate measure of predicted variant function, was variable and heterogeneous in relation to effect size.”
- Figures:

Supplemental Figure 7. Heatmap of bioinformatic tool functional predictions for variants of uncertain significance with large effect sizes

Heatmaps display functional predictions by bioinformatic tools for large effect-size variants of uncertain significance (VUS). Red boxes represent high predicted functional impact, yellow boxes represent low

predicted functional impact, and grey boxes represent no prediction available. Panels A, B, and C display data for rare variants associated with LDL-C, QTc, and HbA1c endophenotypes, respectively. Definitive familial hypercholesterolemia (FH) genes include *LDLR*, *APOB*, *PCSK9*. Definitive long-QT syndrome (LQTS) genes include *KCNQ1*, *KCNH2*, *SCN5A*. Common maturity-onset diabetes of the young (MODY) genes include *HNF1A*, *HNF1B*, *HNF4A*, *GCK*; the color bar indicates the corresponding gene for the variants shown.

Supplemental Figure 8. Heatmap of bioinformatic tool functional predictions for variants with conflicting assertions with large effect sizes

Heatmaps display functional predictions by bioinformatic tools for large effect-size variants with conflicting assertions in ClinVar. Red boxes represent high predicted functional impact, yellow boxes represent low predicted functional impact, and grey boxes represent no prediction available. Panels A, B, and C display data for rare variants associated with LDL-C, QTc, and HbA1c endophenotypes, respectively. Definitive familial hypercholesterolemia (FH) genes include *LDLR*, *APOB*, *PCSK9*. Definitive long-QT syndrome (LQTS) genes include *KCNQ1*, *KCNH2*, *SCN5A*. Common maturity-onset diabetes of the young (MODY) genes include *HNF1A*, *HNF1B*, *HNF4A*, *GCK*; the color bar indicates the corresponding gene for the variants shown.

3. The authors consider rare variants to be those with <1% MAF in disease causing genes. Based on this, the variant set considered for re-classification includes many variants that would not typically be considered via ACMG criteria and may explain why 1) these variants did not feature in ClinVar; and 2) 27.8% of variants were reported to be synonymous (i.e. a surprising result for disease-causing variants). Prior work has demonstrated that allele frequency is important when considering variant classification status and the ClinGen familial hypercholesterolemia Expert Panel suggest the ACMG PM2 rule requires a MAF of <0.02% for *LDLR* (<https://www.medrxiv.org/content/10.1101/2021.03.17.21252755v1>). As such, the current approach documented for variants that do not feature in ClinVar is not appropriate and should either be removed or critically revised to align with community standards.

Author response: We have decreased the rare variant frequency threshold to MAF < 0.1%. The net effect is that the number of P/LP variants have remained the same, while numbers of B/LB/VUS/C variants have decreased. Our findings with regard to carriers of P/LP variants and

P/LP variant effect size compared to other variants remain consistent. We agree that variant frequency is an important consideration for classification status and may differ based on the disease of interest, though it can be difficult to determine appropriate thresholds due to the rarity of some monogenic diseases.

The proportion of synonymous variants with large effect sizes that were not previously submitted to ClinVar and that were synonymous was similar after decreasing the MAF filter to < 0.1% (29.4%), as compared to a MAF filter of < 1% (29.0%). The 29.0% is different from the reviewer's cited 27.8% as we have uncovered an error in our code for variant consequence annotation which has now been fixed. 3.8% of synonymous variants with large effect sizes are located in splice regions, compared to 2.2% of synonymous variants with small effect size (Fisher's exact test p-value = 0.16). Nevertheless, we cannot exclude that synonymous variants with large effect sizes are more likely to confer a true biological impact than synonymous variants without large effect sizes. We also acknowledge that rare coding variants may be subject to imprecise effect size estimation, which we now state in the discussion.

We posit that effect size may be best applied in conjunction with other tools such as prediction algorithms, minor allele frequencies, etc., to screen variants not previously submitted to ClinVar for potential pathogenicity. As such, we have added additional analyses characterizing the functional impact of the variants not submitted to ClinVar with large effect size using *in silico* tools which are included with comment 2.

Revised manuscript:

- Discussion:
 - "We anticipate that a potential application may be to utilize effect size information, when available, either prior to initiating a formal variant classification process or in conjunction with existing methods. Such applications will require prospective evaluation."
 - "We acknowledge that effect size thresholds may differ by disease, given test characteristics that vary according to effect size threshold and endophenotype. Further analyses are warranted to examine the prognostic implications of large-effect variation, test different effect size thresholds for screening potential pathogenicity, and discover easily ascertainable endophenotypes for other monogenic diseases to aid in variant classification."
 - "Rare coding variants may be subject to imprecise effect size estimation which we anticipate will improve with larger repositories of sequence and phenotypic data over time."

References

1. Pirruccello, J. P. *et al.* Deep learning enables genetic analysis of the human thoracic aorta. *Nat Genet* (2021) doi:10.1038/s41588-021-00962-4.
2. Pirruccello, J. P. *et al.* Analysis of cardiac magnetic resonance imaging in 36,000 individuals yields genetic insights into dilated cardiomyopathy. *Nat Commun* **11**, 2254 (2020).
3. Greenwood, T. A. *et al.* Genome-wide Association of Endophenotypes for Schizophrenia From the Consortium on the Genetics of Schizophrenia (COGS) Study. *JAMA Psychiatry* **76**, 1274–1284 (2019).
4. Liu, X., Li, C., Mou, C., Dong, Y. & Tu, Y. dbNSFP v4: a comprehensive database of transcript-specific functional predictions and annotations for human nonsynonymous and splice-site SNVs. *Genome Medicine* **12**, 103 (2020).
5. Tang, H. & Thomas, P. D. Tools for Predicting the Functional Impact of Nonsynonymous Genetic Variation. *Genetics* **203**, 635–647 (2016).
6. Dong, C. *et al.* Comparison and integration of deleteriousness prediction methods for nonsynonymous SNVs in whole exome sequencing studies. *Hum. Mol. Genet.* **24**, 2125–2137 (2015).
7. Choi, S. H. *et al.* Monogenic and Polygenic Contributions to Atrial Fibrillation Risk: Results from a National Biobank. *Circ. Res.* (2019) doi:10.1161/CIRCRESAHA.119.315686.
8. Choi, S. H. *et al.* Rare Coding Variants Associated with Electrocardiographic Intervals Identify Monogenic Arrhythmia Susceptibility Genes: A Multi-ancestry Analysis. *Circ Genom Precis Med* (2021) doi:10.1161/CIRCGEN.120.003300.
9. Conomos, M. P., Miller, M. B. & Thornton, T. A. Robust inference of population structure for ancestry prediction and correction of stratification in the presence of relatedness. *Genet Epidemiol* **39**, 276–293 (2015).
10. Choi, S. H. *et al.* Association Between Titin Loss-of-Function Variants and Early-Onset Atrial Fibrillation. *JAMA* **320**, 2354–2364 (2018).
11. Jurgens, S. J. *et al.* Rare Genetic Variation Underlying Human Diseases and Traits: Results from 200,000 Individuals in the UK Biobank. 2020.11.29.402495
<https://www.biorxiv.org/content/10.1101/2020.11.29.402495v1> (2020) doi:10.1101/2020.11.29.402495.
12. Cardiovascular Disease Knowledge Portal - Home. <https://cvd.hugeamp.org/>.
13. Howe, K. L. *et al.* Ensembl 2021. *Nucleic Acids Res* **49**, D884–D891 (2021).

REVIEWER COMMENTS

Reviewer #1 (Remarks to the Author):

I highly appreciate the careful revision of the manuscript by the authors. Most of my comments have been addressed adequately, a few remain to be answered:

1) It is still unclear how the authors handle the multiple testing situation. The response by the authors is not sufficient in my eyes, because the issue of multiple testing occurs whenever more than one statistical test is being performed. This is at least the case for the results presented in Supplementary Figures 4, 5, 6.

2) The authors now provide confidence intervals as well as estimates of predictive values, which is highly informative. However, a discussion of these values is missing.

3) Further results have been added concerning different estimates of the diagnostic accuracy when applying different thresholds. Given that the interpretation focuses on the proportion of as yet unresolved variants to be classified as pathogenic, at least a statement should be added that these proportions would also differ with different thresholds. Also, I am still missing a discussion on that the performance in currently unresolved variants is unknown, and that these might systematically differ.

4) The statement on the power of quantitative endophenotypes in the introduction was improved. However, still, phenotypes themselves are not more or less powerful, since power is a feature of the statistical approach.

Reviewer #2 (Remarks to the Author):

I thank the authors for their detailed responses to the reviewer queries. There are still some outstanding issues with this manuscript.

"Author response: While there is a small fraction of VUS and conflicting variants which passed the large effect size threshold, there are many more of such variants with negative and/or low effect size which dilute the signal. This pattern remains using a more stringent MAF threshold of < 0.1%."

I would not say that 10-25% of VUS/C that have large effect size is a small fraction. I presume the lack of any signal in the VUS/C carriers is due to the fact that while 10-25% of the distinct variants have large effect sizes, this corresponds to a much lower percentage of carriers (as shown in new Figure 1, lower panel), i.e. the highly recurrent variants in these groups display no phenotypic effect. If this is the case, it should be made clearer in the results, by indicating the proportion of carriers with high effect size variants in addition to the proportion of distinct variants.

For the UKBB variants designated as having large effect sizes, there were 253 ClinVar VUS, 139 ClinVar conflicting and 661 non-ClinVar.

Conflicting variants - 112 missense in total, but 113 when adding the three groups together (71+38+4)?

Non-ClinVar - where does the 807 number come from (Line 198)? Is it a combination of UKBB and replication datasets? This is not very clear.

There remain large numbers of synonymous variants with large effect sizes (22 ClinVar conflicting and 237 non-ClinVar). I do not find the authors explanation of this (a marginal, non-significant enrichment of large effect variants in splice regions, 3.8% compared to 2.2% of small effect size synonymous variants) particularly convincing. This study is arguing that endophenotype effect size can be used to adjudicate pathogenicity - if such a large number of synonymous variants meet the thresholds, they are either having some functional effect (like on splicing) or there is an issue with the approach, thresholds etc. No details are provided for these variants (e.g. in Tables S10/S11), so the result is rather glossed over. In studies such as this, synonymous variants are often used as a "control set" (subject to caveats) where little to no effect on phenotype is observed. It is therefore concerning to see so many with an apparent effect on phenotype.

It would be interesting to assess SpliceAI scores for these variants for example - comparing large effect size variants with small (or non-large) effect size.

It would also be interesting to assess what proportion of non-ClinVar variants for each variant class (missense, synonymous, LOF) had large versus small effect sizes? Do a higher proportion of missense/LOF display large effect sizes?

Finally, one minor error to correct:

"There were 139 non-synonymous variants with conflicting assertions in ClinVar across FH, LQTS, and MODY genes for which large effect size thresholds provided evidence of pathogenicity. Of these, 112

(80.6%) were missense variants, 22 (15.8%) were synonymous variants, 2 (1.4%) were splice donor variants, 2 were frameshift or stop-gained variants (1.4%), and 1 (0.7%) was an in-frame deletion." - Variants cannot be both non-synonymous and synonymous - I would remove the "non-synonymous" description here for the 139 variants.

Reviewer #3 (Remarks to the Author):

Thank you for taking into consideration the previous comments. I am satisfied that these have been appropriately addressed.

We would like to thank the editors and reviewers for their comments, which we feel have strengthened the manuscript.

Reviewer #1 (Remarks to the Author):

I highly appreciate the careful revision of the manuscript by the authors. Most of my comments have been addressed adequately, a few remain to be answered:

1) It is still unclear how the authors handle the multiple testing situation. The response by the authors is not sufficient in my eyes, because the issue of multiple testing occurs whenever more than one statistical test is being performed. This is at least the case for the results presented in Supplementary Figures 4, 5, 6.

Author response: Each reported p-value in Supplemental Figures 4-6 is derived from a single multiple linear regression model. As such, we set the nominal significance at an alpha of 0.05. We do not adjust for multiple testing because the number of variables in the model is extensive, and many can be considered variables that are introduced simply to explain model variance and minimize residual confounding (i.e., age, sex, principal components of ancestry, etc.). Furthermore, we submit that reporting multiple hypothesis adjusted P values only for each of the parameters corresponding to pathogenicity categories included in the multiple linear regression model alone, without multiple hypothesis testing adjustment for other parameters in the model for which a statistical test is inherently performed, would be disingenuous. Therefore, we simply report unadjusted P values, and describe the models. As such, the reader has the opportunity to view the statistical testing (and inference using the 95% confidence intervals provided) in a manner in which they deem appropriate. The details of the multiple linear regression models are shown for each trait below and described in the Methods.

*lm(LDL - C ~ carrier of P variant + carrier of LP variant + carrier of LB variant
+ carrier of VUS + carrier of C variant + age + sex + HDL - C
+ history of myocardial infarction + history of statin usage + PC 1 - 12)*

*lm(QTc ~ carrier of P variant + carrier of LP variant + carrier of LB variant
+ carrier of VUS + carrier of C variant + age + sex
+ history of myocardial infarction + history of heart failure
+ history of beta blocker usage + history of calcium channel blocker usage
+ PC 1 - 12)*

*lm(HbA1c ~ carrier of P variant + carrier of LP variant + carrier of LB variant
+ carrier of VUS + carrier of C variant + age + sex + MCV
+ history of diabetes medication usage + PC 1 - 12)*

Revised and Relevant manuscript text:

- Methods:
 - “We next assessed the estimated difference in individual endophenotype value for carriers of variants in each pathogenicity category relative to carriers of benign (B) variants using a multiple linear regression model. In the model, we regressed individual endophenotype values against carrier status of variants in each pathogenicity category, adjusting for age, sex,

clinical covariates, and first 12 principal components of ancestry. Unadjusted P values corresponding to the pathogenicity category terms were reported.”

2) *The authors now provide confidence intervals as well as estimates of predictive values, which is highly informative. However, a discussion of these values is missing.*

Author response: we have included discussion of these confidence intervals and predictive value estimates to the manuscript.

Relevant and Revised manuscript:

- Results:
 - “As a secondary analysis, we tabulated the sensitivity, specificity, positive predictive value (PPV), and negative predictive value (NPV) of other effect size thresholds based on a range of SD thresholds (Supplemental Table 9). For LDL-C levels, a large effect size threshold corresponding to 0.5 SD of the LDL-C distribution in the UKBB was 16.7 mg/dL. In the UKBB, the large LDL-C effect size threshold had a sensitivity of 76% (95% CI 63, 88) and specificity of 88% (95% CI 85, 91) for discriminating ClinVar designated pathogenic variants from non-pathogenic variants (likely benign, and benign). PPV and NPV were calculated to be 37% (95% CI 27, 47) and 97% (95% CI 96, 99) respectively. The large effect variant threshold in the UKBB provided evidence of pathogenicity for 15.9% of VUS and 18.9% of variants with conflicting assertions in FH genes (Figure 2A). The large effect size threshold corresponding to 0.5 SD of the QTc distribution in the UKBB was 11.9 ms, which had a sensitivity of 80% (95% CI 62, 98) and specificity of 74% (95% CI 69, 79), with PPV of 19% (95% CI 10, 27) and NPV of 98% (95% CI 96, 100). In the UKBB, this threshold provided evidence of pathogenicity for 24.4% of VUS and 16.6% of conflicting variants in the LQTS genes (Figure 2B). For HbA1c, the large effect threshold was 0.31%, with a sensitivity of 70% (95% CI 50, 90), specificity of 95% (95% CI 89, 100), PPV of 82% (95% CI 64, 100), and NPV 90% (95% CI 82, 98); this threshold provided evidence of pathogenicity for 10.4% of VUS and 8.8% of conflicting variants in the MODY genes (Figure 2C).”
- Discussion:
 - “We acknowledge that effect size thresholds that provide evidence of pathogenicity may differ by endophenotype and disease. Moreover, as with most tests the appropriateness of a given effect size threshold may vary by intended use of the information, which may justify a more sensitive or specific threshold (e.g., screening individuals for potential monogenic disease risk, clinical reporting of variant pathogenicity, etc.). Greater precision in effect size threshold test characteristics will follow from larger repositories of sequence and phenotype data in the future. Further analyses are warranted to examine the prognostic implications of large-effect variation, test different effect size thresholds for screening potential pathogenicity, and discover easily ascertainable endophenotypes for other monogenic diseases to aid in variant classification.”

3) *Further results have been added concerning different estimates of the diagnostic accuracy when applying different thresholds. Given that the interpretation focuses on the proportion of as yet unresolved variants to be classified as pathogenic, at least a statement should be added that these proportions would also differ with different thresholds. Also, I am still missing a discussion on that the performance in currently unresolved variants is unknown, and that these might systematically differ.*

Author response: we have included additional discussion addressing these points.

Relevant and Revised manuscript:

- Discussion: "The large number of novel variants discovered with sequencing efforts highlights a need for rapid and scalable methods for assessing variant pathogenicity. Indeed, many individuals in our studies carried rare coding variants in disease susceptibility genes that have not previously been classified or submitted to ClinVar with clinical pathogenicity assertions (58.7% of FH variants, 30.0% of LQTS variants, and 81.5% of MODY variants in the UKBB). We acknowledge that effect size thresholds that provide evidence of pathogenicity may differ by endophenotype and disease. Moreover, as with most tests the appropriateness of a given effect size threshold may vary by intended use of the information, which may justify a more sensitive or specific threshold (e.g., screening individuals for potential monogenic disease risk, clinical reporting of variant pathogenicity, etc.). Greater precision in effect size threshold test characteristics will follow from larger repositories of sequence and phenotype data in the future."

4) *The statement on the power of quantitative endophenotypes in the introduction was improved. However, still, phenotypes themselves are not more or less powerful, since power is a feature of the statistical approach.*

Author response: we have modified this statement accordingly.

Revised manuscript:

- Introduction: "Quantitative endophenotypes are instrumental phenotypes for genetic association and are often more easily and reliably ascertained than dichotomous disease status indicators."

Reviewer #2 (Remarks to the Author):

I thank the authors for their detailed responses to the reviewer queries. There are still some outstanding issues with this manuscript.

"Author response: While there is a small fraction of VUS and conflicting variants which passed the large effect size threshold, there are many more of such variants with negative and/or low effect size which dilute the signal. This pattern remains using a more stringent MAF threshold of < 0.1%."

I would not say that 10-25% of VUS/C that have large effect size is a small fraction. I presume the lack of any signal in the VUS/C carriers is due to the fact that while 10-25% of the distinct variants have large effect sizes, this corresponds to a much lower percentage of carriers (as

shown in new Figure 1, lower panel), i.e. the highly recurrent variants in these groups display no phenotypic effect. If this is the case, it should be made clearer in the results, by indicating the proportion of carriers with high effect size variants in addition to the proportion of distinct variants.

Author response: We have provided a table in the supplement tabulating the proportion of VUS or conflicting variants with a large effect size, as well as the proportion of individuals that carry them. We note that in FOURIER, a clinical trial enriched for patients with atherosclerotic cardiovascular disease, we observe a substantial enrichment in the proportion of carriers of VUS or conflicting assertion with large effect sizes. We submit this observation is in keeping with a clinically relevant effect of such variants on the LDL and HbA1c endophenotypes.

Revised manuscript:

- Methods: “The percentage of carriers of variants of large effect size with ClinVar uncertain significance or conflicting assertion was calculated for each endophenotype and cohort.”
- Results: “Within the UKBB, the fraction of carriers of VUS or conflicting assertion variants with large effect size varied by endophenotype (LDL 4.7%, QTc 11.3%, and HbA1c 1.5%). In contrast, in FOURIER, a clinical trial enriched for patients with atherosclerotic cardiovascular disease, we observed an increase in the percentage of carriers of VUS or conflicting assertion variants with large effect sizes for the LDL (16.0%) and HbA1c (11.0%) endophenotypes (Supplemental Table 11). We submit that the increased fraction of carriers of VUS or conflicting assertion variants with large effect size for the LDL and HbA1c endophenotypes in FOURIER is consistent with a functional and clinically impactful role of such variants.”
- Supplement:

Supplemental Table 11. Variants with large effect sizes observed in ClinVar with uncertain significance or conflicting assertions

Endophenotype	Cohort	Large effect size variants, n (%)	Carriers of large effect size variants, n (%)
LDL	UKBB	155 (16.9)	703 (4.7)
	FOURIER	59 (19.4)	113 (16.0)
QTc	UKBB	96 (21.4)	207 (11.3)
	TOPMed	95 (19.0)	250 (12.1)
HbA1c	UKBB	10 (9.9)	39 (1.5)
	FOURIER	7 (24.1)	11 (11.0)

Large effect size refers to variants with effect size > 0.5 standard deviations of the endophenotype distribution in the UK Biobank. Variants observed in ClinVar with uncertain significance (VUS) or conflicting assertions (C) are included in this table.

For the UKBB variants designated as having large effect sizes, there were 253 ClinVar VUS, 139 ClinVar conflicting and 661 non-ClinVar.

Conflicting variants - 112 missense in total, but 113 when adding the three groups together (71+38+4)?

Author response: We have confirmed that there are 112 missense variants total, but 113 variants with predicted functional impact (PFI) score available including 2 splice-donor variants and 1 stop-gained variant. The text has been updated with this clarification.

Revised manuscript:

- Results: “The PFI score was calculated for 71 variants in genes associated with FH with a median PFI of 0.63 (Q1-Q3 0.25-0.77), with the highest PFI observed in LDLR variants (n=53, median PFI 0.76, Q1-Q3 0.57-0.80). There were 38 variants in genes associated with LQTS with a median PFI of 0.66 (Q1-Q3 0.33-0.82) and 4 variants in genes associated with MODY with a median PFI of 0.80 (Q1-Q3 0.66-0.86).”

Non-ClinVar - where does the 807 number come from (Line 198)? Is it a combination of UKBB and replication datasets? This is not very clear.

Author response: There were 807 unique, large effect size variants identified spanning all cohorts studied (UKBB and replication datasets). The text has been updated with this clarification.

Revised manuscript:

- Results: “In total, 807 unique large effect size variants were identified across all cohorts, of which 544 (67.4%) were missense, 237 (29.4%) synonymous, 22 (2.7%) were loss-of-function (LOF; frameshift, stop-gained, or splice-altering), and the remainder with various consequences.”

There remain large numbers of synonymous variants with large effect sizes (22 ClinVar conflicting and 237 non-ClinVar). I do not find the authors explanation of this (a marginal, non-significant enrichment of large effect variants in splice regions, 3.8% compared to 2.2% of small effect size synonymous variants) particularly convincing. This study is arguing that endophenotype effect size can be used to adjudicate pathogenicity - if such a large number of synonymous variants meet the thresholds, they are either having some functional effect (like on splicing) or there is an issue with the approach, thresholds etc. No details are provided for these variants (e.g. in Tables S10/S11), so the result is rather glossed over. In studies such as this, synonymous variants are often used as a "control set" (subject to caveats) where little to no effect on phenotype is observed. It is therefore concerning to see so many with an apparent effect on phenotype.

It would also be interesting to assess what proportion of non-ClinVar variants for each variant class (missense, synonymous, LOF) had large versus small effect sizes? Do a higher proportion of missense/LOF display large effect sizes?

Author response: We have examined the proportion of large effect size variants stratified by variant consequence, including loss of function (frameshift, stop-gained, and splice altering variants), missense and indels, and synonymous variants. As expected, a higher proportion of loss of function variants have large effect sizes compared to missense/indels and synonymous variants. The pattern is similar with a larger effect size threshold (1 SD of the endophenotype distribution). We have now included

Supplemental Figure 13 to demonstrate the pattern, and we also highlight variants not specifically in ClinVar. We agree with the reviewer that future analysis of the potential

functional role of synonymous variants with large effect sizes is warranted. We also report the number of large and small effect size variants by variant classification and consequence in **Supplemental Table 13**.

Revised manuscript:

- Methods: “Then, we examined the proportion of large effect size variants stratified by variant consequence, including loss-of-function (frameshift, stop-gained, and splice altering variants), missense and indels, and synonymous variants.”
- Results: “As expected, we observed that the fraction of variants with large effect sizes was greatest for loss-of-function variants (38%), followed by missense/indels (22%), and synonymous variants (18%; Supplemental Table 13, Supplemental Figure 9). The pattern of large effect size variants stratified by variant consequence was similar when we used an effect size threshold of one standard deviation of the endophenotype distribution.”
- Discussion: “As expected, a higher proportion of loss-of-function variants had large effect sizes compared to missense/indels and synonymous variants. Similar proportions were observed for missense/indels and synonymous variants, which we submit likely reflects substantial variability in the functional role of such variants.”
- Supplement:

Supplemental Table 13. Variants with large effect size stratified by variant consequence

SD threshold	0.5						1.0					
Variant consequence	Loss of function, n		Missense and indel, n		Synonymous, n		Loss of function, n		Missense and indel, n		Synonymous, n	
Effect size	Large	Small	Large	Small	Large	Small	Large	Small	Large	Small	Large	Small
Not in ClinVar	22	91	551	1,961	15	98	15	98	263	2,222	101	985
Overall	79	128	1,054	3,728	60	147	60	147	517	4,248	182	2,105

Loss of function includes frameshift, stop-gained, and splice altering variants. Large effect size refers to variants with effect size > 0.5 standard deviations of the endophenotype distribution in the UK Biobank.

Supplemental Figure 9. Proportion of large effect size variants by variant consequence

Loss of function includes frameshift, stop-gained, and splice altering variants. Large effect size refers to variants with effect size > 0.5 standard deviations of the endophenotype distribution in the UK Biobank.

It would be interesting to assess SpliceAI scores for these variants for example - comparing large effect size variants with small (or non-large) effect size.

Author response: We have conducted this analysis and included relevant data below. There were a small number of synonymous variants identified to have increased probability of being splice-altering; the sparsity of data limited the ability to draw conclusions about whether large effect size synonymous variants are more likely to be splice-altering than small effect size variants.

Revised Manuscript:

- Methods: “Synonymous variants located in genes associated with each endophenotype were analyzed using SpliceAI for splice-altering potential.¹ Delta scores of 0-1 were generated, with 1 representing the highest probability of the variant being splice-altering.”
- Results: “A total of 2,261 unique synonymous variants were analyzed using SpliceAI, with 68 (3.0%) of these demonstrating some probability of being splice-altering with delta score > 0.2 (Supplemental Table 14).”
- Supplement:

Supplemental Table 14. Splice-altering predictions of synonymous variants

SpliceAI score	Delta score ≥ 0.2 , n		Delta score ≥ 0.5 , n		Delta score ≥ 0.8 , n	
	Large	Small	Large	Small	Large	Small
Not in ClinVar	5	25	1	6	0	3
Overall	14	54	6	14	1	5

Suggested SpliceAI cutoffs in order of increased probability of splice-altering effect are 0.2 (high recall), 0.5 (recommended), and 0.8 (high precision). Large effect size refers to variants with effect size > 0.5 standard deviations of the endophenotype distribution in the UK Biobank.

Finally, one minor error to correct:

"There were 139 non-synonymous variants with conflicting assertions in ClinVar across FH, LQTS, and MODY genes for which large effect size thresholds provided evidence of pathogenicity. Of these, 112 (80.6%) were missense variants, 22 (15.8%) were synonymous variants, 2 (1.4%) were splice donor variants, 2 were frameshift or stop-gained variants (1.4%), and 1 (0.7%) was an in-frame deletion." - Variants cannot be both non-synonymous and synonymous - I would remove the "non-synonymous" description here for the 139 variants.

Author response: we have removed the “non-synonymous” description accordingly.

Reviewer #3 (Remarks to the Author):

Thank you for taking into consideration the previous comments. I am satisfied that these have been appropriately addressed.

Author response: we thank the reviewer for their comments.

References Cited

1. Jaganathan, K. et al. Predicting Splicing from Primary Sequence with Deep Learning. Cell 176, 535-548.e24 (2019).

REVIEWERS' COMMENTS

Reviewer #1 (Remarks to the Author):

I have no further comments, my previous concerns have all been resolved.

Reviewer #2 (Remarks to the Author):

I thank the authors for their detailed responses to reviewer queries. I have no further comments and congratulate the authors for their work on this manuscript.